# Causal role of the angular gyrus in insight-driven memory reconfiguration

**Anna-Maria Grob[1], Hendrik Heinbockel[1], Branka Milivojevic[2], Christian F Doeller[3,4,5], Lars Schwabe[1]***

[1]Department of Cognitive Psychology, Institute of Psychology, Universität Hamburg, Hamburg, Germany; [2]Radboud University, Donders Institute for Brain, Cognition and Behaviour, Nijmegen, Netherlands; [3]Kavli Institute for Systems Neuroscience, Centre for Neural Computation, The Egil and Pauline Braathen and Fred Kavli Centre for Cortical Microcircuits, Jebsen Centre for Alzheimer's Disease, Norwegian University of Science and Technology, Trondheim, Norway; [4]Max-Planck-Insitute for Human Cognitive and Brain Sciences, Leipzig, Germany; [5]Wilhelm Wundt Institute of Psychology, Leipzig University, Leipzig, Germany

*For correspondence:
lars.schwabe@uni-hamburg.de

Competing interest: The authors declare that no competing interests exist.

**Abstract** Maintaining an accurate model of the world relies on our ability to update memory representations in light of new information. Previous research on the integration of new information into memory mainly focused on the hippocampus. Here, we hypothesized that the angular gyrus, known to be involved in episodic memory and imagination, plays a pivotal role in the insight-driven reconfiguration of memory representations. To test this hypothesis, participants received continuous theta burst stimulation (cTBS) over the left angular gyrus or sham stimulation before gaining insight into the relationship between previously separate life-like animated events in a narrative-insight task. During this task, participants also underwent EEG recording and their memory for linked and non-linked events was assessed shortly thereafter. Our results show that cTBS to the angular gyrus decreased memory for the linking events and reduced the memory advantage for linked relative to non-linked events. At the neural level, cTBS targeting the angular gyrus reduced centro-temporal coupling with frontal regions and abolished insight-induced neural representational changes for events linked via imagination, indicating impaired memory reconfiguration. Further, the cTBS group showed representational changes for non-linked events that resembled the patterns observed in the sham group for the linked events, suggesting failed pruning of the narrative in memory. Together, our findings demonstrate a causal role of the left angular gyrus in insight-related memory reconfigurations.

## eLife assessment

This **important** paper provides **solid** evidence that the angular gyrus plays a role in insight-based memory updating. The study is well conducted, timely, and presents clear-cut behavioral results. While the study provides robust evidence that transcranial magnetic stimulation to the angular gyrus impacts memory, evidence for the strong claim of a causal contribution of the angular gyrus in particular – apart from other connected regions, including the hippocampus – is not conclusive.

## Introduction

The capacity to flexibly update our memories in light of new information is fundamental to maintaining an accurate model of the world around us. This flexibility requires adaptable memory networks that can be reconfigured upon acquiring new insights. Previous research provided direct evidence for

insight-induced reconfigurations of memory representations and showed that insight into the connection of initially separate events propels the integration of these events into coherent episodes (*Collin et al., 2015*; *Milivojevic et al., 2015*). Such mnemonic integration allows novel inferences (*Spalding et al., 2018*; *Zeithamova et al., 2012*) that aid efficient navigation (*Coutanche et al., 2013*; *Fernandez et al., 2023*; *He et al., 2022*) and decision-making (*Boorman et al., 2021*; *Kumaran et al., 2009*; *Shohamy and Daw, 2015*). Importantly, in everyday life, the inference about the relationship between seemingly unrelated events is often not inferred via direct observation but through imagination. For instance, when reading a book, we gain insight into the plot and possible twists through our imagination, which then prompts us to update our memory representations. Even when new insights are derived from direct observation, the integration process requires imaginative capacities to bind the previously separate memories into a coherent narrative. At the neural level, the hippocampus has been shown to play a pivotal role in (imagination-based) mnemonic integration (*Cohn-Sheehy et al., 2021a*; *Collin et al., 2015*; *Griffiths and Fuentemilla, 2020*; *Grob et al., 2023a*; *Milivojevic et al., 2015*). However, while the hippocampus appears to be crucial for mnemonic integration, it does not act in isolation but operates in collaboration with cortical areas to accomplish this complex process (*Backus et al., 2016*; *Milivojevic et al., 2015*; *Pehrs et al., 2018*; *Schlichting and Preston, 2015*; *Spalding et al., 2018*). Yet, our understanding of the specific areas implicated in the insight-driven reconfiguration of memory representations, beyond the hippocampus, remains limited. Moreover, existing data on the neural underpinnings of mnemonic integration are mainly correlational in nature and which areas are causally involved in the integration of initially unrelated memories into cohesive representations is completely unknown.

One promising candidate that may contribute to insight-driven memory reconfiguration is the angular gyrus. The angular gyrus has extensive structural and functional connections to many other brain regions (*Petit et al., 2023*), including the hippocampus (*Coughlan et al., 2023*; *Uddin et al., 2010*). Accordingly, previous studies have shown that stimulation of the angular gyrus resulted in altered hippocampal activity (*Thakral et al., 2020*; *Wang et al., 2014*). Furthermore, the angular gyrus has been implicated in a myriad of cognitive functions, including mental arithmetic, visuospatial processing, inhibitory control, and theory-of-mind (*Cattaneo et al., 2009*; *Grabner et al., 2009*; *Lewis et al., 2019*; *Schurz et al., 2014*). Moreover, there is accumulating evidence pointing to a key role of the angular gyrus in long-term memory (*Bellana et al., 2017*; *Bonnici et al., 2018*; *Kwon et al., 2022*; *Wang et al., 2014*) and imagination (*Ramanan et al., 2018*; *Thakral et al., 2017*; *Thakral et al., 2020*). How these putative functions of the angular gyrus relate to one another, however, remained unclear. We reasoned that these functions might be directly linked, enabling the angular gyrus to drive the integration of (imagination-related) insights into long-term memory. In line with this idea, recent theories propose that the angular gyrus acts as dynamic buffer for spatiotemporal representations (*Humphreys et al., 2021*), which may allow the angular gyrus to transiently maintain the initially separate events and to integrate these into cohesive narratives. This buffering function of the angular gyrus may be particularly relevant for imagination-based linking. Thus, we hypothesized that the angular gyrus plays a crucial role in integrating imagination-related insights into long-term memory and hence in the dynamic reconfiguration of memory representations in light of new information.

To test this hypothesis and determine the causal role of this area in insight-related memory reconfigurations, we conducted a preregistered study combining a life-like video-based narrative-insight task (*Milivojevic et al., 2015*; *Figure 1*), probing insight-related reconfigurations of memory, with representational similarity analysis of EEG data and (double-blind) 'neuro-navigated' TMS to an area of the left angular gyrus that was implicated in imaginative processing before (*Thakral et al., 2017*). Considering this involvement of the angular gyrus in imaginative processes, we expected that the effect of cTBS on the change in representational similarity from pre- to post-insight will differ based on the mode of insight – whether this insight was gained via imagination or observation. Specifically, we expected a more pronounced impairment in the neural reconfigurations when insight is gained via imagination, as this function may depend more on angular gyrus recruitment than insight gained via observation. Additionally, we expected cTBS to the left angular gyrus to reduce the increase in neural similarity for linked events and increase of neural dissimilarity for non-linked events. We further predicted that cTBS to the left angular gyrus would specifically reduce the impact of (imagination-based) insight into the link of initially unrelated events on memory performance during free recall,

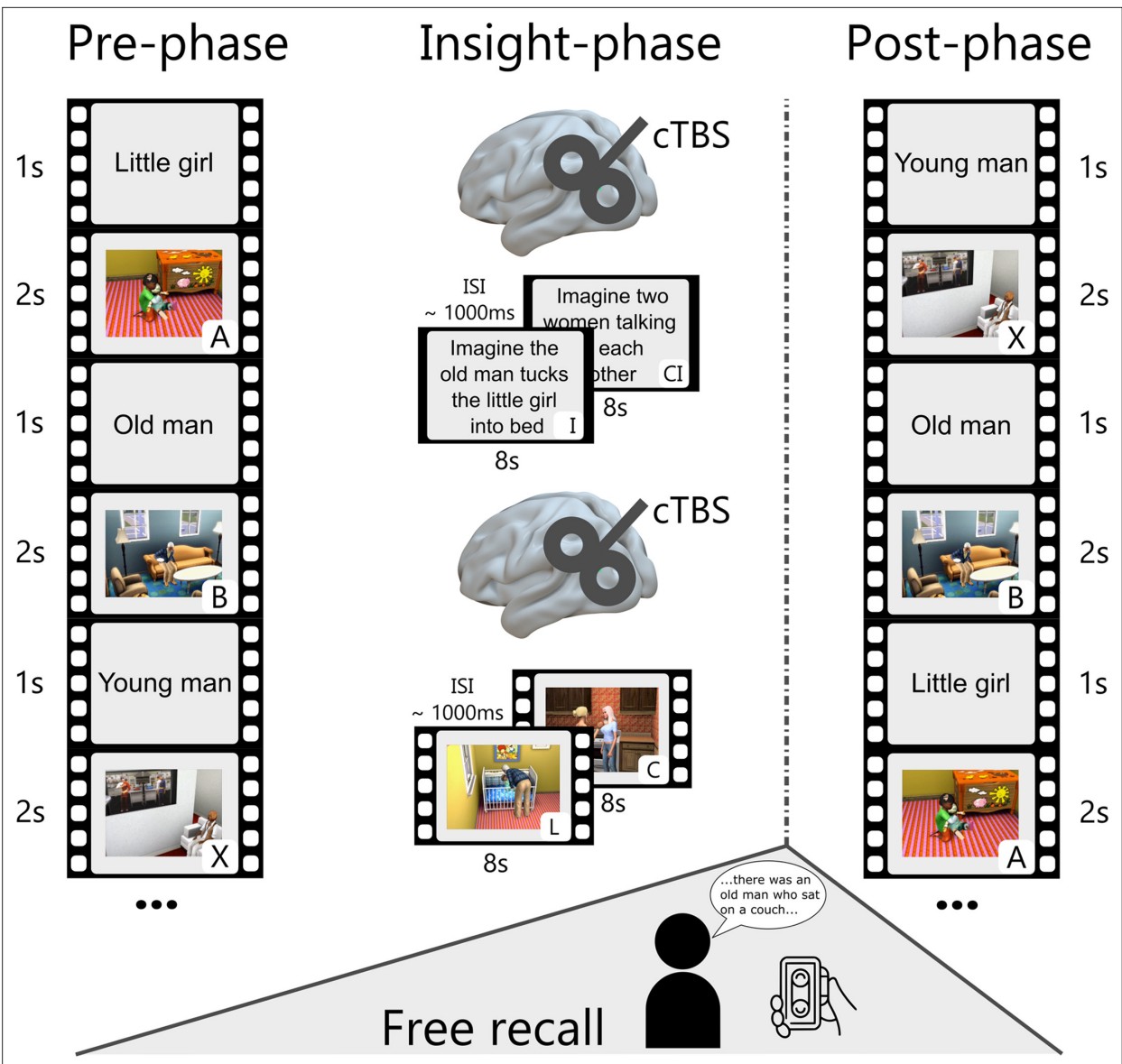

**Figure 1.** Modified narrative-insight task and procedure. During the pre-phase, participants viewed video events (**A, B**, and **X**) from 10 different storylines. Each event was preceded by a title (1 s) and repeated 18 times. The inter-stimulus interval (ISI) was ~1000ms. The subsequent insight-phase consisted of two parts. In one part, participants gained insight through a written imagination instruction (**I**) interspersed with a control instruction (**CI**). In the other part, they gained insight through a linking video (**L**) interspersed with a control video (**C**). The order of gaining insight through imagination or video observation was counterbalanced across participants. Before each insight part, participants received, depending on the experimental group, either a sham or cTBS stimulation over the left angular gyrus (MNI: −48,−67, 30). After the insight-phase, participants had a 30-min break and then completed a free recall for a maximum of 20 min in a different room. In the post-phase, all video events were presented in the same manner as the pre-phase.

given the higher variability of free recall compared to other memory measures with lower search demands. Considering the high connectivity profile of the angular gyrus within the brain (*Seghier, 2013*), we conducted an EEG connectivity analysis building upon findings from the RSA analyses concerning alterations in neural reconfigurations. To establish a link between neural and behavioral findings, we chose a correlational approach to relate observations from these two domains. We intentionally adopted a mixed design, combining both between-subjects and within-subject methodologies. The between-subjects approach was chosen to minimize the risk of carry-over effects and sequence biases. Simultaneously, we capitalized on the advantages of a within-subject design by altering the pre- to post-insight comparison and the mode of insight (imagination vs. observation)

**Table 1.** Control variables.

| Measure | Sham | | cTBS | | |
| --- | --- | --- | --- | --- | --- |
| | $M$ | $SD$ | $M$ | $SD$ | $p_{uncorr}$ |
| FFIS-C | 24.06 | 4.24 | 23.59 | 4.64 | 0.066 |
| FFIS-D | 16.84 | 5.42 | 17.86 | 5.22 | 0.446 |
| FFIS-E | 13.84 | 7.16 | 15.00 | 7.10 | 0.514 |
| FFIS-F | 27.87 | 8.96 | 28.21 | 9.63 | 0.885 |
| STAI-T | 34.13 | 9.27 | 38.62 | 11.15 | 0.082 |
| STAI-S | 35.35 | 7.60 | 39.65 | 10.08 | 0.056 |
| TICS | 11.97 | 8.54 | 13.41 | 9.95 | 0.531 |
| BDI | 6.84 | 6.88 | 7.65 | 7.57 | 0.654 |
| BFI-2 E | 43.10 | 5.66 | 40.59 | 8.54 | 0.165 |
| BFI-2 N | 27.13 | 6.94 | 30.12 | 10.09 | 0.166 |
| BFI-2 O | 47.16 | 6.99 | 46.68 | 6.89 | 0.779 |
| BFI-2 C | 40.42 | 8.06 | 40.71 | 6.78 | 0.878 |
| BFI-2 A | 48.94 | 5.06 | 46.74 | 5.76 | 0.106 |
| Age | 25.45 | 4.62 | 23.62 | 3.82 | 0.088 |
| MT | 53.03 | 14.59 | 54.82 | 12.93 | 0.608 |

Note. The questionnaires FFIS with its dimensions: FFIS-C (complexity of imagination), FFIS-D (directedness of imagination), FFIS-E (emotional valence of imagination), FFIS-F (frequency of imagination); STAI-T and STAI-S; TICS; BDI; BFI-2 with its dimensions: BFI-2 E (extraversion), BFI-2 N (neuroticism), BFI-2 O (openness to experience), BFI-2 C (conscientiousness), BFI-2 A (agreeableness) were completed during the 30-min break after the insight-phase. Age in years. Motor thresholds (MT) in percent of maximum stimulator capacity. No significant group differences were observed on any of these measures. p Values are displayed uncorrected for multiple comparisons. Data represents means (+/-$SD$).

within each participant. To control for any group differences beyond the TMS manipulation, we gathered various control variables through questionnaires, including trait- and state-anxiety, depressive symptoms, chronic stress levels, personality dimensions, and imaginative capacities.

## Results

### cTBS to the angular gyrus reduces insight-related memory boost

The angular gyrus has been implicated in a myriad of tasks and functions, including long-term memory (*Bonnici et al., 2018*; *Kwon et al., 2022*; *Wang et al., 2014*) and imagination (*Ramanan et al., 2018*; *Thakral et al., 2017*). Here, we hypothesized that these functions of the angular gyrus are directly linked to one another. Specifically, we postulated that the angular gyrus plays a crucial role in the integration of imagination-related insights into long-term memory representations and that it thus represents a key player in the dynamic reconfiguration of memory in light of new information. To test this hypothesis and the causal role of the angular gyrus in insight-related memory reconfigurations, we combined the life-like video-based narrative-insight task with representational similarity analysis of EEG data and (double-blind) neuro-navigated TMS over the left angular gyrus in a comprehensive investigation within a single day. During the narrative-insight task, participants first saw three video events (A, B, and X; pre-phase), which were then either linked into a narrative (A and B) or not (A and X) in a subsequent insight-phase. Critically, before the insight-phase, we applied either sham stimulation (31 participants, 15 females) or continuous theta burst stimulation (cTBS; 34 participants, 16 females) to the left angular gyrus. Notably, the groups did not differ on levels of subjective chronic stress (TICS), state and trait anxiety (STAI-S, STAI-T), depressive mood (BDI), imaginative capacities (FFIS), personality dimensions (BFI), age, and motor thresholds (for descriptive statistics see *Table 1*; all p>0.056).

Following the insight-phase and a 30-min break to mitigate potential TMS aftereffects (*Huang et al., 2005*; *Jannati et al., 2023*), participants completed a free recall task, which provided a measure of insight-related changes in subsequent memory. Thereafter, participants saw the same video events (A, B, and X) again in a post-phase. EEG was measured during all stages of the narrative-insight task. Contrasting neural representation patterns from the pre- and post-phases allowed us to assess insight-related memory reconfiguration and its modulation by cTBS to the angular gyrus. Due to its specific relevance in imaginative processes (*Ramanan et al., 2018*; *Thakral et al., 2017*; *Thakral et al., 2020*), we expected that the angular gyrus would be particularly relevant if insight relies strongly on imagination. Therefore, participants gained insight into half of the stories by imagining the link themselves, while they observed the link as a video in the other half of the stories. Participants' ratings showed that they adhered well to these instructions during the linking phase. When linking events via imagination, they reported imagining the linking events very well ($M = 3.38$, $SD = 0.47$) and their imagination as depictive ($M = 3.35$, $SD = 0.46$). When linking via observation, they reported a high level of understanding of the linking events ($M = 3.37$, $SD = 0.51$) and found the linking events meaningful ($M = 3.35$, $SD = 0.52$) on a 1–4 Likert scale. Furthermore, participants demonstrated a high level of attention throughout the narrative-insight task, responding to target stimuli with near-ceiling performance ($M = 99.25\%$; $SD = 1.40 \%$) without any group differences ($t(63.00) = 0.42$, $p = 0.675$, $d = -0.10$).

Importantly, participants were unaware of the allocation to the cTBS or sham condition, as indicated by the treatment guess at the end of the experiment (Fisher's exact test; $p = 0.597$). Furthermore, TMS stimulation did not affect participants' subjective mood, wakefulness or arousal (mood: *group ×time*: $F(1, 63) = 0.76$, $p = 0.386$, $\eta_G G 0.00$; wakefulness: *group ×time*: $F(1, 63) = 0.01$, $p = 0.921$, $\eta_G G 0.00$; arousal: *group ×time*: $F(1, 63) = 0.01$, $p = 0.921$, $\eta_G G 0.00$).

As expected, all participants gained insight into which events were linked in the narrative-insight task, as they rated the belongingness of linked events higher than non-linked events from pre- to post-insight, as indicated by a linear mixed model (LMM: *time ×link*: $\beta = 2.49$, 95% CI [2.15, 2.83], $t(418.44) = 14.01$, $p<0.001$; *Figure 2—figure supplement 1*). Post-hoc tests showed increasing belongingness ratings for linked events and decreasing belongingness ratings for non-linked events from pre- to post-insight (LMM: *link*: $\beta = 1.49$, 95% CI [1.33, 1.64], $t(418) = 24.49$, $p<0.001$; *non-link*: $\beta = -.91$, 95% CI [-1.07, -.76], $t(418.00) = -15.03$, $p<0.001$). This insight was further reflected in the multi-arrangements task (MAT), in which participants were instructed to arrange representative images (A, B, and X) from each story based on their relatedness. In this task, all participants arranged linked events closer together than non-linked events (MAT; LMM: *link*: $\beta = -1.33$, 95% CI [-1.59,-1.07], $t(177.00) = -9.81$, $p<0.001$; *Figure 2—figure supplement 2*). The strong insight gained by all participants was further reflected in their near-ceiling performance in the forced-choice recognition task, in which participants were instructed to identify the event (B or X) that was linked with A. Participants accurately indicated whether B or X was linked to A (sham: $M = 94.65\%$, $SD = 9.00\%$; cTBS: $M = 97.34\%$, $SD = 6.10\%$; *Figure 2—figure supplement 3*). Importantly, there were no group differences in any of these measures (LMM: narrative-insight task: *group × time × link*: $\beta = -0.02$, 95% CI [-0.49, 0.45], $t(418.27) = -0.09$, $p = 0.929$; LMM: multi-arrangements task: *group × link*: $\beta = 0.11$, 95% CI [-0.26, 0.48], $t(177.00) = 0.58$, $p = 0.561$; LMM: Forced-choice recognition: *group*: $\beta = 0.23$, 95% CI [-0.26, 0.72], $t(113.37) = 0.90$, $p = 0.368$), indicating that all participants successfully gained insight into which events were linked and that the (left) angular gyrus did not play a critical role in the process of gaining insight itself.

To investigate the causal role of the left angular gyrus in insight-related episodic memory integration, the key question of this study, we first analyzed the detailedness of participants' memory for both linked and non-linked events during free recall. Across groups, linked events were generally recalled in more detail than non-linked events (LMM: *link*: $\beta = 1.20$, 95% CI [0.86, 1.54], $t(406.00) = 6.75$, $p<0.001$), suggesting a memory boost for integrated narratives. Most interestingly, cTBS to the left angular gyrus through cTBS reduced this insight-related memory boost for linked events significantly (LMM: *group × link*: $\beta = -0.54$, 95% CI [-1.02,-0.06], $t(406.00) = -2.17$, $p = 0.030$; *Figure 2A*). Pairwise comparisons revealed a significantly lower number of recalled details for linked events in the cTBS compared to the sham group, while there was no significant difference for non-linked events (LMM: *link*: $\beta = -0.40$, 95% CI [-0.79,-0.02], $t(85.50) = -2.79$, $p = 0.033$; LMM: *non-link*: $\beta = -0.09$, 95% CI [-0.29, 0.42], $t(406.00) = -0.62$, $p = 0.926$). Additionally, we observed that all participants showed better memory for central compared to peripheral details of the plot when recalling linked events,

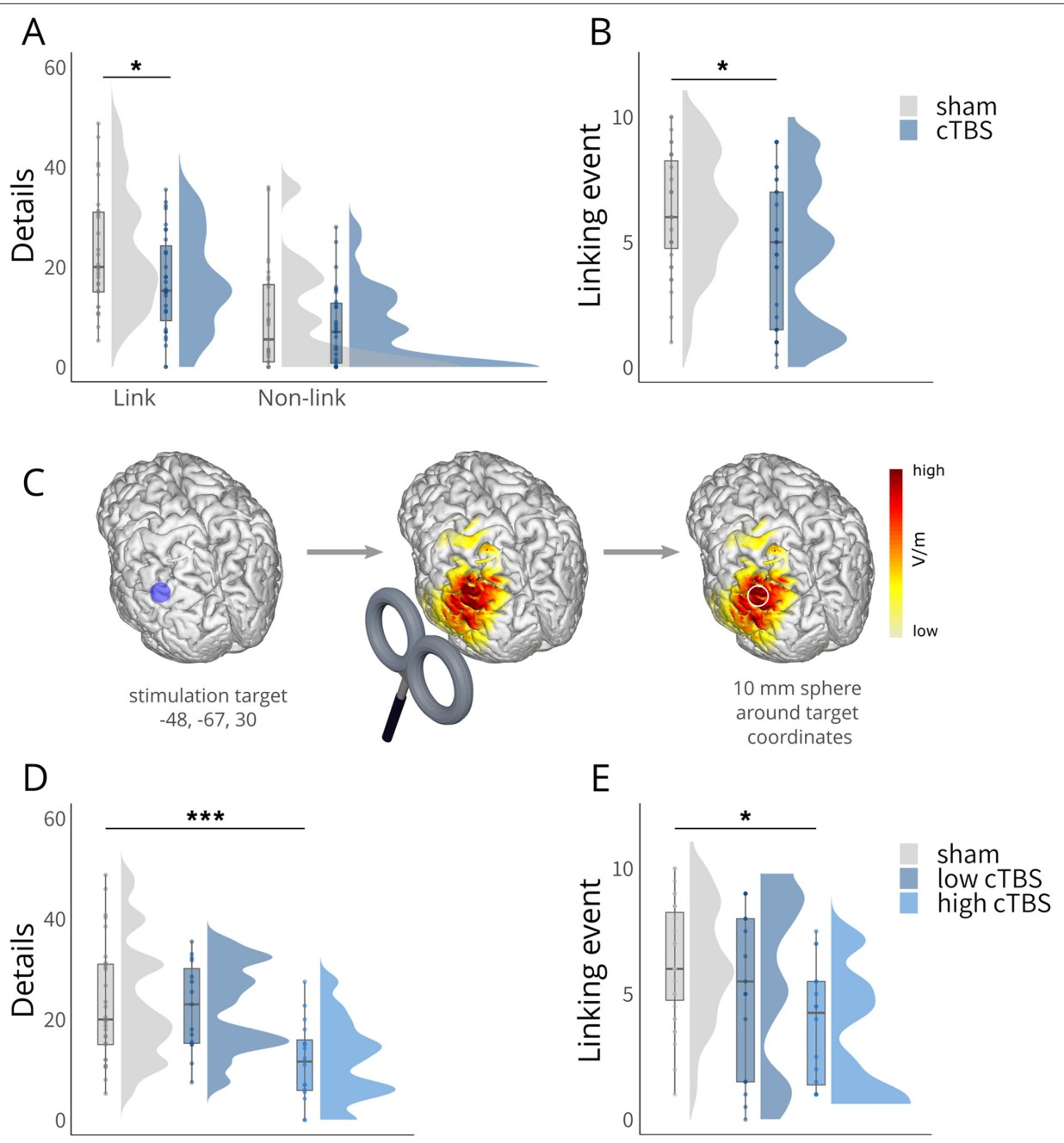

**Figure 2.** Behavioral results. (**A**) Significantly reduced recall of details for linked events in the cTBS group compared to the sham group, with no significant difference for the non-linked events. (**B**) Significantly reduced recall of the linking events in the cTBS group compared to the sham group. (**C**) Schematic overview of electric field modeling: Simulation was performed for the angular gyrus coordinate (MNI: x = –48, y = –67, z = 30) using a Magstim 70 mm figure-of eight coil at 80% of individual motor thresholds, reflecting the applied setup. The resulting electric field was averaged within a 10 mm spherical ROI and centered on the target coordinate and extracted for subsequent analyses. Please note that in the study, the coil handle was oriented upwards; however, in this illustration, it has been intentionally depicted as pointing downwards for better visibility purposes. (**D**) Significantly reduced number of details recalled for linked events specifically in the high cTBS group (based on a median-split on simulated electric field strengths). (**E**) Significantly reduced recall of the linking events specifically in the high cTBS group (based on a median-split on simulated electric field strengths). Boxplots show the median for each group. Boxplot whiskers extend to the minimum or maximum value within 1.5 times the interquartile range. Points within the boxplot indicate individual data points per each group. Density plots indicate data distribution per group. The belongingness ratings for the linked and non-linked events are shown in *Figure 2—figure supplement 1*, the data of the multiple arrangements task in *Figure 2—figure supplement 2*, and the data of the forced-choice recognition test in *Figure 2—figure supplement 3*. Statistical differences stem from pairwise post-hoc tests of marginal means. *$p<0.05$, ***$p<0.001$.

*Figure 2 continued on next page*

*Figure 2 continued*
The online version of this article includes the following figure supplement(s) for figure 2:
**Figure supplement 1.** Results of the narrative-insight task (NIT).
**Figure supplement 2.** Results of the multiple-arrangements task (MAT).
**Figure supplement 3.** Results of the forced-choice recognition task.

which was not observed to the same extent for non-linked events (LMM: *link × detail: β* = 0.61, 95% *CI* [0.12, 1.09], *t*(406.00) = 2.43, p = 0.016).

In a second step, we analyzed whether cTBS to the angular gyrus affected, in addition to memory detailedness for initially separate but now linked events, also the memory for the linking events themselves. Our results showed that cTBS (vs. sham) significantly reduced the frequency with which participants recalled the linking events (LMM: *group: β* = –0.66, 95% *CI* [-1.13,–0.18], *t*(98.13) = –2.71, p = 0.008; *Figure 2B*). Interestingly, this TMS effect appeared to be particularly pronounced when events were linked via imagination (cTBS vs. sham: *t*(61.46) = –2.53, p = 0.014, *d* = –0.63) and was less prominent when they were linked via direct observation (cTBS vs. sham: *t*(58.59) = - 1.63, p = 0.107, *d* = –0.40), although it is important to note that the interaction was not significant (LMM: *group × mode: β* = 0.30, 95% *CI* [–0.18, 0.77], *t*(62) = 1.23, p = 0.225).

To assess the effect of cTBS stimulation on the angular gyrus (*Pizem et al., 2022*; *Zhang et al., 2022*), we performed electric field simulations at 80% of the individual motor threshold, averaging the estimated field strength within a 10 mm sphere centered around the angular gyrus coordinate (MNI: −48,–67, 30). In order to examine whether the behavioral effects were dependent on the simulated electric field strength (*Figure 2C*), we next included electric field strength (strong vs. weak via median split) and repeated the previous linear mixed model predicting the number of details for linked events including a group factor reflecting stimulation strength (sham, low, high). This model yielded a significant *group × link* interaction (LMM: *β* = –0.78, 95% *CI* [-1.35,–0.21,], *t*(399.00) = –2.63, p = 0.009; *Figure 2D*), suggesting a dependency of memory on stimulation strength. Pairwise comparisons for linked events confirmed that a stronger electric field induction in the angular gyrus significantly reduced the memory boost for linked events, while there was no such effect for weak cTBS stimulation (LMM: sham vs. low: *β* = 0.05, 95% *CI* [–0.45, 0.55], *t*(87.70) = 0.28, p = 1.000; sham vs. high: *β* = 0.74, 95% *CI* [0.25, 1.23], *t*(87.7) = 4.45, *p*<0.001, low vs. high: *β* = 0.69, 95% *CI* [0.13, 1.26], *t*(87.7) = 3.60, p = 0.007). We further included the electric field strength (strong vs. weak via median split) and repeated the previous linear mixed model predicting the naming of the linking events including the group factor stimulation strength (sham, low, high). This analysis yielded a significant effect of *group* (LMM: *β* = –0.92, 95% *CI* [-1.50,–0.35], *t*(97.79) = –3.11, p = 0.003; *Figure 2E*), suggesting that the memory for the linking events was dependent on the angular gyrus stimulation strength.

## Angular gyrus stimulation disrupts neural pattern reconfiguration following imagination-based insight

Our behavioral data showed that cTBS to the angular gyrus reduced the insight-related memory boost. In a next step, we tested whether cTBS to the angular gyrus may also alter the insight-related reconfiguration of neural memory representations, taking the mode of insight (i.e. imagination vs. observation) into account. To this end, we leveraged representational similarity analysis (RSA) of EEG data and compared changes in multivariate oscillatory theta power patterns for linked and non-linked events from pre- to post-insight (*Figure 3A*). We focused exclusively on the theta band since theta has been shown to hold a key role in episodic memory integration (*Backus et al., 2016*; *Nicolás et al., 2021*). For this analysis, similarity maps (time × time) were computed by correlating story-specific theta frequency patterns within linked (A with B) and within non-linked (A with X) events in the pre- and post-phase, separately. Subsequently, we examined insight-induced effects on neural representations for linked (vs. non-linked) events by comparing the change from pre- to post-insight (post-pre) and the difference between imagination and observation (imagination - observation) between cTBS and sham groups using an independent cluster-based permutation t-test.

First, we included the within-subject factors time (pre vs. post), mode of insight (imagination vs. observation) and link (vs. non-link) by calculating the difference waves. Subsequently we conducted a cluster-based permutation test comparing the cTBS and the sham groups. This analysis yielded a

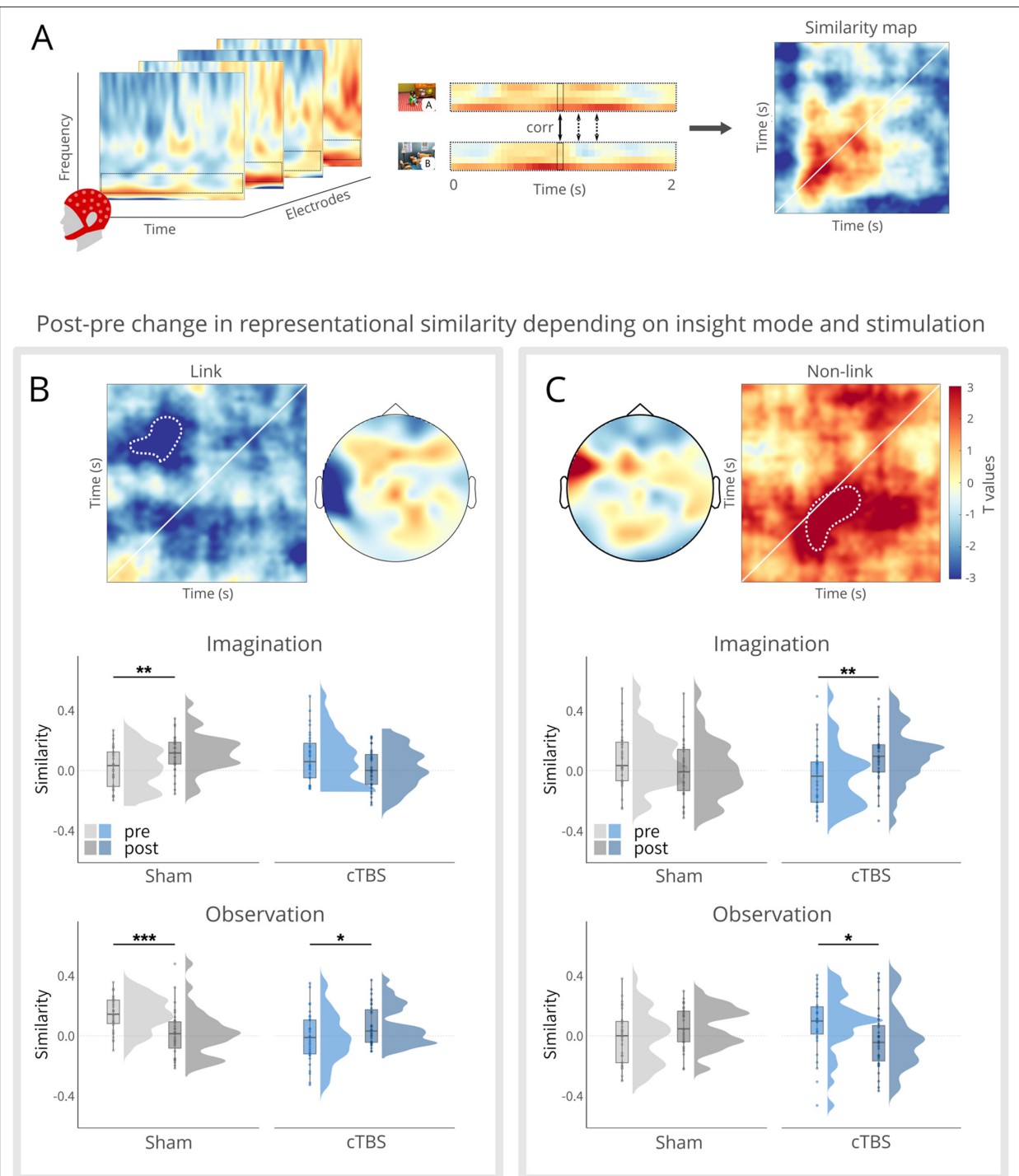

**Figure 3.** Representational pattern changes. (**A**), Conceptual overview of the representational similarity analysis (RSA) on theta oscillations. First, time-frequency data was computed, and the theta power values (4–7 Hz) were extracted. Using these feature vectors, Pearson's correlations were computed to compare the power patterns across time points of events (here: event A and B). These correlations resulted in a time × time similarity map. (**B**), Significant cluster, denoted by white dotted line for illustrative purposes, for the change from post-pre and imagination-observation between the cTBS and sham groups using an independent sample cluster-based permutation t-test for linked events (A and B). In the middle panel, follow-up tests on stories linked via imagination revealed increased similarity for the sham group, while no significant effect was observed for the cTBS group. In the lower panel, follow-up tests on stories linked via observation showed decreased similarity for the sham group and increased similarity for the cTBS group. (**C**), Significant cluster, denoted by white dotted line for illustrative purposes, for the change from post-pre and imagination-observation between the cTBS and sham groups using an independent sample cluster-based permutation t-test for non-linked events (A and X). In the middle panel, follow-

*Figure 3 continued on next page*

*Figure 3 continued*

up tests on stories linked via imagination revealed increased similarity for the cTBS group, while no significant effect was observed for the sham group. In the lower panel, follow-up tests on stories linked via observation showed decreased similarity for the cTBS group and no significant effect for the sham cTBS group. Boxplots show the median similarity for each group at each time point. Boxplot whiskers extend to the minimum or maximum value within 1.5 times the interquartile range. Points within the boxplot indicate individual data points in each group. Density plots indicate data distribution per group and time. *$p<0.05$, **$p<0.01$, ***$p<0.001$.

The online version of this article includes the following figure supplement(s) for figure 3:

**Figure supplement 1.** Raw representational dissimilarity maps for linked events.

**Figure supplement 2.** Raw representational dissimilarity maps for non-linked events.

four-way interaction within a negative cluster in a fronto-temporal region (electrode: FT7; p = 0.007, *ci*-range = 0.00, SD = 0.00). This result indicates that the impact of cTBS over the angular gyrus on the neural pattern reconfiguration following imagination- vs. observation-based insight may differ between linked and non-linked events. For linked events, this analysis yielded a negative cluster (p = 0.032, *ci*-range = 0.00, SD = 0.00) in the parieto-temporal region (electrodes: T7, Tp7, P7; *Figure 3B*; *Figure 3—figure supplement 1*). Follow-up tests on the extracted similarity cluster analyzed the representational pattern change and its modulation by TMS separately for the imagination and observation condition. For stories linked via imagination, we obtained an increase in representational similarity from pre- to post-insight in the sham group ($t(30) = 3.48$, p = 0.002, $d_{repeated\ measures} = 0.62$), whereas there was no such increase and even a trend for a decrease in representational similarity for linked events from pre- to post-insight in the cTBS group ($t(30) = -2.01$, p = 0.053, $d_{repeated\ measures} = -0.36$; *group × time*: $F(1, 60) = 14.03$, $p<0.001$, $\eta_G = 0.09$; *Figure 3B* middle panel). Interestingly, we observed that a lower change (post - pre) in representational similarity of events linked via imagination (vs. observation) was associated, across groups, with a reduced probability of recall of the linking events ($r = 0.27$, $t(59) = 2.17$, p = 0.034), suggesting a direct association between neural pattern reconfiguration and subsequent memory. Furthermore, to address a deviation from the normality assumption, the correlational analysis was repeated using the Spearman method, which indicated a stronger correlation ($r(59) = 0.32$, p = 0.012).

For stories that were linked via observation, we observed a seemingly opposite pattern (*group × time*: $F(1, 60) = 19.21$, $p<0.001$, $\eta_G = 0.12$): decreased similarity in the sham group ($t(30) = -3.94$, $p<0.001$, $d_{repeated\ measures} = -0.62$) but increased representational similarity in the cTBS group ($t(30) = 2.30$, p = 0.029, $d_{repeated\ measures} = 0.62$; *Figure 3B* lower panel). However, these changes in representational similarity for the observation condition should be interpreted with caution, as these seemingly opposite changes appeared to be at least in part driven by group differences already in the pre-phase, before participants gained insight.

Interestingly, we observed a different pattern of insight-related representational pattern changes for non-linked events. Similarly to linked events, we compared the change from pre- to post-insight and the difference between imagination and observation between cTBS and sham using an independent sample cluster-based permutation t-test. This analysis yielded a positive cluster (p = 0.035, *ci*-range = 0.00, SD = 0.00) in a fronto-temporal region (electrode: FT7; *Figure 3C*; *Figure 3—figure supplement 2*). Again, we pursued this effect with separate follow-up tests for the imagination and observation conditions. In the imagination condition, the sham group did not show any representational changes for non-linked events ($t(30) = -1.35$, p = 0.187; $d_{repeated\ measures} = -0.23$), while we observed increased neural similarity for non-linked events from pre- to post-insight in the cTBS group ($t(30) = 3.61$, p = 0.001, $d_{repeated\ measures} = 0.67$; *Figure 3C* middle panel). Thus, participants who received cTBS to the angular gyrus showed a pattern of pre- to post-insight representational changes for non-linked events that resembled the pattern observed in the sham group for events linked via imagination, suggesting that cTBS to the angular gyrus before gaining imagination-based insight interfered with efficient pruning of the integrated narrative.

For stories linked via observation, we observed, again, a seemingly opposite pattern (*group × time*: $F(1, 60) = 10.32$, p = 0.002, $\eta_G = 0.07$): no representational change in the sham group ($t(30) = 1.65$, p = 0.110, $d_{repeated\ measures} = 0.34$) but decreased neural similarity for non-linked events from pre- to post-insight in the cTBS group ($t(30) = -2.40$, p = 0.023, $d_{repeated\ measures} = -0.42$). Again, these representational changes should be interpreted with caution, as the differences appear to be at least in part driven by group differences in the pre-insight phase.

## cTBS to the angular gyrus diminishes fronto-temporal connectivity associated with imagination-based insight

To further elucidate the neural mechanisms involved in the changes in insight-related memory reconfiguration after cTBS to the left angular gyrus, we next examined changes in functional connectivity using the same contrast as in the previous RSA analyses. More specifically, we computed imaginary coherence for the mean theta frequency using a sliding window approach and tested the change in connectivity from pre- to post-insight for linking via imagination vs. observation between the cTBS and the sham groups using an independent sample cluster-based permutation t-test.

This analysis yielded a negative cluster (p = 0.044, $ci$-range = 0.00, SD = 0.00; time window: 1.25–1.75 s; *Figure 4A*) between centro-temporal and frontal regions (C4 – Fp1; C6 – Fp1; T8- Fp1; T8 – AF7). For stories linked via imagination, follow-up tests indicated decreased functional connectivity between these regions in the cTBS group ($t(30)$ = –4.25, $p<0.001$, $d_{repeated\ measures}$ = –0.70), while there was no change in the sham group ($t(30)$ = 0.02, p = 0.987, $d_{repeated\ measures}$ = 0.00; group × time: $F(1, 60)$ = 8.05, p = 0.006, $\eta_G$G0.050; *Figure 4B*). Interestingly, across both groups higher coherence between these areas in the post- relative to the pre-phase for stories linked via imagination relative to observation was associated with better recall of details (central and peripheral) for stories linked via imagination ($r$ = 0.31, $t(59)$ = 2.50, p = 0.015; *Figure 4C*), suggesting that the reduced crosstalk between these regions was linked to impaired subsequent memory. Please note that for addressing a deviation from the normality assumption, the correlational analysis was repeated using the Spearman method, which yielded an significant correlation of similar strength ($r(59)$ = 0.31, p = 0.015).

For stories linked via observation, we found a seemingly opposite pattern (group × time: $F(1, 60)$ = 12.73, $p<0.001$, $\eta_G$ = .080; *Figure 4D*): decreased functional connectivity for the sham group ($t(30)$ = –2.75, p = 0.010, $d_{repeated\ measures}$ = –0.47), while the cTBS group exhibited increased connectivity ($t(30)$ = 2.28, p = 0.030, $d_{repeated\ measures}$ = 0.36). Since these differences appeared to be again already present prior to gaining insight, the functional connectivity changes obtained for the observation condition should be interpreted with caution. Regarding non-linked events (X), we did not find any significant cluster in this coherence analysis (all clusters $p>0.221$), indicating that the reported connectivity changes were specific to linked events.

## Control variables

Overall, levels of subjective chronic stress, anxiety, and depressive mood were relatively low and not different between groups. The groups did further not differ in terms of personality traits, imagination capacity, age or motor thresholds (all $p>0.056$; see *Table 1*).

## Discussion

Updating our memory representations in light of new information is key to keeping an accurate model of the world. Given the previously described role of the angular gyrus in episodic memory and imagination (*Benoit and Schacter, 2015*; *Bonnici et al., 2018*; *Thakral et al., 2017*), we hypothesized that this area plays a pivotal role in (imagination-based) insight-driven memory reconfiguration. To probe the causal role of the angular gyrus in insight-induced mnemonic changes and related neural pattern reconfigurations, we combined a life-like narrative insight task with cTBS to the angular gyrus. Our results show that cTBS to the angular gyrus reduced the insight-related memory boost for linked events and decreased memory for the linking events themselves. At the neural level, cTBS to the angular gyrus reduced the coupling between centro-temporal and frontal regions and abolished insight-induced neural representational changes for events linked via imagination, which points to impaired memory integration.

Unsurprisingly, all participants successfully gained insight into which events were linked in the narrative-insight task, as reflected in their reliable recognition of the linked events in the forced-choice recognition task and their relatedness ratings in the multi-arrangements task. Importantly this insight had a direct impact on memory, with better memory for linked relative to non-linked events. This memory boost for linked events aligns with previous findings (*Cohn-Sheehy al., 2021b*; *Grob et al., 2023a*; *Grob et al., 2023b*; *Wang et al., 2015*) and supports the notion that the brain stores episodic memories as coherent narratives (*Tulving, 1983*), for which any element can cue the entire episode (*Horner et al., 2015*; *Nakazawa et al., 2002*). Crucially, while cTBS to the angular gyrus had

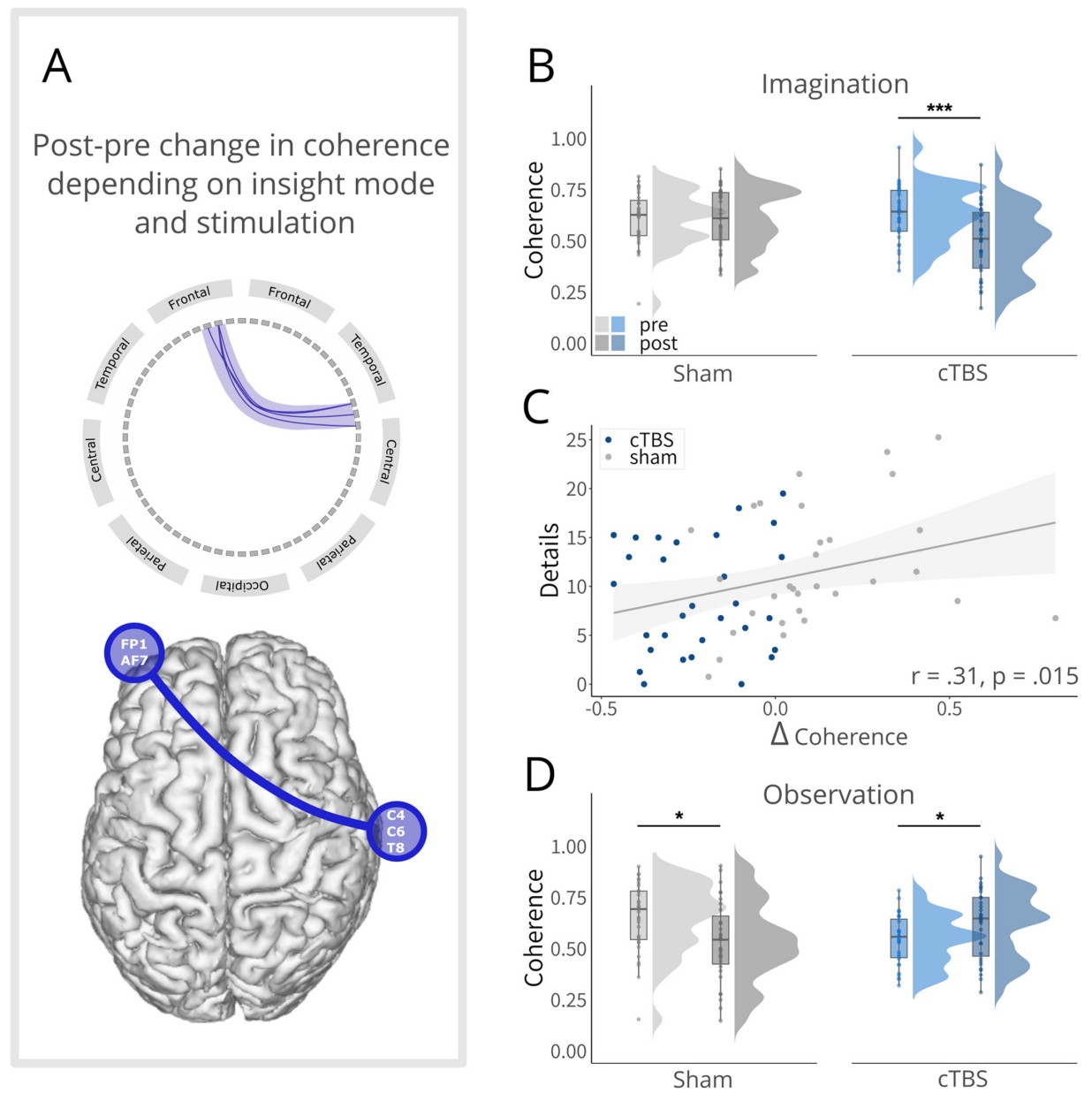

**Figure 4.** Connectivity change for linked events. (**A**), Conceptual depiction of the negative cluster (centro-temporal and frontal). In the upper panel, the connections are presented separately for each electrode pair. The lower panel illustrates the cluster coherence based on the electrode locations relative to the brain. (**B**), Follow-up tests for events linked via imagination indicated decreased coherence between centro-temporal and frontal electrodes for the cTBS group, while no change was observed for the sham group. (**C**), Significant positive correlation, indicating that the less the coherence between centro-temporal and frontal electrodes changed from pre to post for imagination (vs. observation), the fewer details for linked events via imagination were recalled. Please note, that this correlation became even stronger when the outlier was removed ($r = 0.38$, $t(58) = 3.11$, $p = 0.003$). (**D**), Follow-up tests for events linked via observation indicated decreased coherence between centro-temporal and frontal electrodes for the sham group, while the cTBS group showed a significant increase in coherence. Boxplots show the median coherence for each group at each time point. Boxplot whiskers extend to the minimum or maximum value within 1.5 times the interquartile range. Points within the boxplot indicate individual data points in each group. Density plots indicate data distribution per group and time. *$p < 0.05$, **$p < 0.01$, ***$p < 0.001$.

no effect on basic insight, cTBS to the angular gyrus prior to gaining insight into the relationship of previously separate events specifically reduced this memory boost for linked events, particularly when electric field stimulation of the angular gyrus was strong. The diminished memory boost for linked events following effective cTBS to the angular gyrus demonstrates a causal involvement of the angular

gyrus in prioritizing linked narratives in memory and, by implication, the reconfiguration of memories in light of new information.

CTBS to the angular gyrus prior to gaining insight impaired memory not only for the linked events but also for the linking events themselves. Again, this was exclusively observed for cTBS stimulation with a strong electric field, ensuring effective stimulation of the angular gyrus. The resulting impairment in memory for linking events following cTBS to the angular gyrus highlights its causal role in integrating newly acquired information into pre-existing memory representations, suggesting that the linking events – whether observed or imagined – serve as binding information for creating a coherent narrative. This interpretation is in line with a recent study showing that online TMS over the angular gyrus during a reading-based task impaired the integration of contextual information (*Branzi et al., 2021*). Interestingly, recent work has additionally indicated that targeting parietal regions with TMS led to alterations in hippocampal functional connectivity, thereby enhancing associative memory (*Nilakantan et al., 2017*; *Tambini et al., 2018*; *Wang et al., 2014*), potentially shedding light on the underlying mechanisms involved. Dovetailing with recent accounts proposing the angular gyrus as a spatiotemporal buffering region for integrating our continuous stream of experiences (*Humphreys et al., 2021*), cTBS to the angular gyrus may have disrupted the reactivation and maintenance of initially separate events A and B during the linking phase. This disruption may have impeded the binding process that is likely crucial for the observed memory boost. At the same time, the lacking reactivation and integration of previously separate events A and B during the linking phase may have diminished the significance of the linking event, leading to reduced memory for the linking event itself.

Consistent with previous neuroimaging studies demonstrating distinct representational changes associated with insight into the relationship of initially separate events (*Collin et al., 2015*; *Milivojevic et al., 2015*), we observed, in addition to insight-related memory changes, increased similarity in neural representations of events linked via imagination in the theta band from pre- to post-insight. This finding aligns with studies linking theta band activity to hippocampal involvement in memory integration (*Backus et al., 2016*; *Herweg et al., 2020*; *Nicolás et al., 2021*). Mechanistically, hippocampal theta oscillations may facilitate memory integration by promoting more accurate representations of stimulus-specific information while gaining insight (*Pacheco Estefan et al., 2021*). Critically, cTBS to the angular gyrus abolished these insight-induced changes in neural pattern similarity between linked events in the theta band, which was directly associated with reduced recall of the linking events. Beyond its causal involvement in the insight-driven neural reconfiguration of linked event representations, cTBS to the angular gyrus led to representational changes for non-linked events that closely resembled the changes observed in the sham group for linked events. This finding indicates that cTBS to the angular gyrus impairs the separation of linked and non-linked events, suggesting its causal role in effectively pruning out events that are not part of the integrated narrative.

Given the consistent findings linking the angular gyrus to scene imagination (*Addis et al., 2007*; *Hassabis et al., 2007*; *Ramanan et al., 2018*; *Thakral et al., 2017*), our study focused on the role of the angular gyrus in imagination-based mnemonic integration. It is important to note that the neural findings discussed thus far exclusively pertain to imagination-based insight, aligning with the primary focus of our study. When gaining insight via imagination, participants had to retrieve memory representations of initially separate events to construct the linking event in their mind (*Schacter et al., 2008*). We assume that the retrieval and reinstatement necessary to construct the imagined linking event led to a high degree of co-activation of the neural patterns associated with each event, leading to more similar representations (*Wammes et al., 2022*). Most interestingly, cTBS to the angular gyrus' buffering function (*Humphreys et al., 2021*) disrupted the observed increase in similarity for events linked via imagination, lending further support to its causal role in imagination-based memory integration. Together, our findings demonstrate that cTBS to the angular gyrus disrupts neural pattern reconfiguration following imagination-based insight.

Although we specifically targeted the left angular gyrus and identified its causal role in insight-driven memory reconfiguration, the angular gyrus does not act in isolation. As a major connector hub, it integrates information from various brain regions, with connections including the prefrontal and parietal regions (*Frey et al., 2008*; *Makris et al., 2005*; *Makris et al., 2007*), striatum (*Petit et al., 2023*; *Uddin et al., 2010*), sensory-motor areas (*Bonner et al., 2013*), and the medial-temporal lobe, including the hippocampus (*Uddin et al., 2010*). Some of these regions play a key role in

insight-induced memory reconfigurations, in particular the hippocampus and prefrontal cortex (*Collin et al., 2015*; *Milivojevic et al., 2015*). Consistent with this, cTBS to the angular gyrus resulted in decreased theta band connectivity between centro-temporal and frontal regions for events linked via imagination. Importantly, this reduction in connectivity was directly related to a subsequent memory decline. This finding, specific to linked events, aligns with prior research indicating that stronger (and inter-hemispheric) functional coupling in the theta band between frontal and posterior regions is linked to enhanced associative memory (*Cruzat et al., 2021*; *Summerfield and Mangels, 2005*; *Wu et al., 2007*). Furthermore, in line with our finding that is specific to imagination-based linking, the importance of theta band synchronization has been highlighted in research focused on mental imagery (*Li et al., 2009*). These results suggest that by disrupting the angular gyrus, long-range theta synchronization for events linked via imagination is also disrupted, directly impacting memory performance for these events. This underscores the role of the angular gyrus as a connector hub, integrating information from various regions, particularly during the process of imagination-based memory integration.

To compare the angular gyrus' involvement in imagination-based linking with observation-based linking, where mental construction of linking events is not required, participants linked half of the stories by observing the linking events as videos. For events linked via observation, we found decreased pattern similarity in the theta band between linked events from the pre- to the post-phase. It may be tempting to speculate that viewing the linking event, reactivated only elements of the memory representations of A and B, resulting in a moderate co-activation of their memory representations, which typically leads to decreased similarity between events (*Wammes et al., 2022*) and more distinctive memory representations in line with previous studies (*Grob et al., 2023a*; *Grob et al., 2023b*; *Hein-bockel et al., 2022*). CTBS to the angular gyrus disrupted this decrease, potentially preventing full access to the original memory representation of A and B in all their detailed distinctiveness during the observation of the linking event, subsequently failing to induce a decrease in similarity (*Ramanan et al., 2018*). Following cTBS to the angular gyrus, we further observed decreased pattern similarity for non-linked events in the observation-based condition, resembling the pattern change observed in the sham group for linked events, which may highlight the role of the angular gyrus in representational separation during observation-based linking. Furthermore, for events linked via observation, we observed decreased theta coherence in the sham group. It may be speculated that decreased theta synchronization for events linked via observation indicated less working memory demands in line with prior research (*Fell and Axmacher, 2011*; *Kawasaki et al., 2014*; *Sarnthein et al., 1998*). Importantly, cTBS to the angular gyrus led to increased functional connectivity between centro-temporal and frontal regions, potentially indicating increased working memory demands. However, caution is warranted when interpreting these findings for observation-based insight because these appeared to be driven at least in part by group differences already in the pre-phase, that is before participants' gained insight and before the TMS manipulation.

Although our study provided evidence suggesting a causal role of the angular gyrus in insight-driven memory reconfigurations – highlighted by behavioral changes after cTBS to the angular gyrus, neural changes in left parietal regions, and relevant brain-behavior associations – it is important to acknowledge the limitations imposed by the spatial resolution of EEG. Consequently, the precise source of the observed signal changes in the parietal regions remains uncertain, potentially tempering the definitive nature of these findings. Furthermore, the differential impact of cTBS to the angular gyrus on neural reconfigurations between events linked via imagination and those linked via observation may be attributed to its crucial role in imaginative processes (*Ramanan et al., 2018*; *Thakral et al., 2017*). Another intriguing aspect to consider is that the stimulated site was situated in the more ventral portion of the angular gyrus, recognized for its stronger connectivity to the episodic hippocampal memory system in contrast to its more dorsal counterpart (*Seghier, 2013*; *Uddin et al., 2010*). This stronger connectivity between the ventral angular gyrus and the hippocampus may shed light on the greater impact of cTBS to the angular gyrus on imagination-based insight. Given the angular gyrus's robust connectivity with other brain regions, including the hippocampus (*Seghier, 2013*), it is plausible that the observed changes might not solely stem from alterations within the angular gyrus itself, but could also originate from these interconnected regions. This notion may bear particular importance given the required accessibility to the hippocampus during imaginative processes (*Benoit and Schacter, 2015*; *Grob et al., 2023a*; *Zeidman and Maguire, 2016*). Interactions between the angular gyrus and the hippocampus may give rise to rich memory representations (*Ramanan et al., 2018*). In

line with this, recent studies have demonstrated that cTBS to the angular gyrus resulted in enhanced hippocampal connectivity and improved associative memory (*Hermiller et al., 2019*; *Tambini et al., 2018*; *Wang et al., 2014*). However, it should be noted that our study detected impaired associative memory following cTBS to the angular gyrus. Expanding upon this idea, it is conceivable that targeting a more dorsal segment of the angular gyrus might exert a stronger influence on observation-based linking – an aspect that warrants future investigations. Yet, while acknowledging the functional heterogeneity within the angular gyrus (*Humphreys et al., 2020*), pinpointing specific sub regions via TMS remains challenging due to its limited focal precision at the millimeter level (*Deng et al., 2013*; *Thielscher and Kammer, 2004*), as reinforced by our electric field simulations utilizing a 10 mm radius. Hence, drawing definitive conclusions regarding distinct angular gyrus sub regions requires future research employing rigorous checks to assess the focality of their stimulation.

In conclusion, our data point to a causal involvement of the angular gyrus in (imagination-based) insight-driven memory reconfiguration. These results provide novel insights into the neural mechanisms of memory integration and bridge the traditionally separate functions attributed to the angular gyrus, namely memory and imagination. Beyond their relevance for understanding fundamental memory processes, these findings may have relevant implications for promoting the integration of fragmented memories in mental disorders, such as posttraumatic stress disorder.

## Materials and methods

**Key resources table**

| Reagent type (species) or resource | Designation | Source or reference | Identifiers | Additional information |
|---|---|---|---|---|
| Software, algorithm | Psychophysics Toolbox | Brainard, D. H., & Vision, S. (1997). The psychophysics toolbox. *Spatial vision*, *10*(4), 433–436. | RRID: SCR_002881 | |
| Software, algorithm | MATLAB | The MathWorks Inc, Natick, Massachussetts, USA | RRID: SCR_002881 | |
| Software, algorithm | ActiView | BioSemi B.V., Amsterdam, the Netherlands | RRID: SCR_023671 | |
| Software, algorithm | PowerMag View! System | MAG & More GmbH, Munich, Germany | RRID: SCR_023670 | |
| Software, algorithm | FieldTrip | *Oostenveld et al., 2011* | RRID: SCR_004849 | |
| Software, algorithm | SimNIBS 4 | Thielscher, A., Antunes, A., & Saturnino, G. B. (2015, August). Field modeling for transcranial magnetic stimulation: a useful tool to understand the physiological effects of TMS?. In *2015 37th annual international conference of the IEEE engineering in medicine and biology society (EMBC)* (pp. 222–225). IEEE, https://doi.org/10.1109/EMBC.2015.7318340 | RRID: SCR_014109 | |
| Software, algorithm | Nonparametric cluster-based statistical testing of EEG data | *Maris and Oostenveld, 2007* | | |
| Software, algorithm | R version 4.04 | R Core Team (2014). R: A language and environment for statistical computing. R Foundation for Statistical Computing, Vienna, Austria. URL http://www.R-project.org/ | RRID:SCR_001905 | |

## Preregistration

This study was preregistered before the start of data collection at the German Clinical Trials Register (DRKS-ID: DRKS00025202; https://drks.de/search/de/trial/DRKS00025202). For translation to English, please adjust the page settings located in the top right corner.

## Participants

Sixty-five healthy right-handed individuals (34 males, 31 females, age: *M* = 24.49 years, SD = 4.29 years) with normal or corrected-to-normal vision volunteered to participate in this study. Participants were screened using a standardized interview for exclusion criteria that comprised a history of neurological and psychiatric disease, medication use and substance abuse, cardiovascular, thyroid, or renal disease, evidence of COVID-19 infection or exposure, and any contraindications to MRI examination or TMS. Additionally, participants with a body mass index (BMI) below 19 or above 26 kg/m² were excluded. This decision stemmed from recruiting some participants from prior studies that incorporated stress induction protocols, which imposed this specific criterion (*Herhaus and Petrowski, 2018*; *Schmalbach et al., 2020*). All participants gave written informed consent before participation and received a monetary compensation at the end of the experiment. The procedures were approved by the local ethics committee (Faculty of Psychology and Human Movement Science, Universität Hamburg, Hamburg, Germany, 2020_301 Grob Schwabe) and adhered to the Declaration of Helsinki. The sample size is in line with recent studies on episodic memory integration using the same task (*Grob et al., 2023a*; *Grob et al., 2023b*). Additionally, an a-priori power calculation using G*Power (*Faul et al., 2007*) indicates that a sample size of N = 54 is sufficient for detecting a medium-sized group × link effect (*f* = 0.25) with a power of 0.95.

We implemented a mixed-design including the within-subject factors link (linked vs. non-linked events), session (pre- vs. post-link), and mode (imagination vs. observation) as well as the between-subjects factor group (cTBS to the angular gyrus vs. sham) to mitigate the risk of carry-over effects and sequence biases of the crucial cTBS manipulation. Participants were pseudo-randomly assigned to the cTBS group (n = 34, 16 females) and the sham group (n = 31, 15 females) to achieve a comparable distribution of men and women in each group. Due to technical issues, three cTBS participants were excluded from EEG analyses.

## Procedure

After obtaining participants' written informed consent, we determined their individual motor thresholds for transcranial magnetic stimulation (TMS). Thereafter, they completed a training session of the modified narrative-insight task (*Milivojevic et al., 2015*), a life-like video-based task that tests the integration of initially separate events into coherent episodes (see below). During this training, participants were equipped with electroencephalography (EEG) caps and electrodes. Following the training session, participants completed the pre-phase of the narrative-insight task. After completing a German mood questionnaire (MDBF; *Steyer et al., 1997*), participants underwent either sham or cTBS targeting the left angular gyrus before commencing the insight-phase, with an additional stimulation session administered before the second half of the insight-phase. Crucially, this study was double-blind, ensuring that both the participant and the experimenter were unaware of the stimulation condition. Upon completion of the insight-phase, participants transitioned to another experimental room where they were given a 30-min break, during which they completed the German mood questionnaire (MDBF) again, along with assessments of their imagination capacity (FFIS; *Zabelina and Condon, 2020*), trait-anxiety (STAI-T; *Laux et al., 1981*), and state-anxiety (STAI-S; *Laux et al., 1981*), depressive symptoms (BDI; *Hautzinger et al., 2006*), chronic stress (TICS; *Schulz and Schlotz, 1999*), and personality dimensions (BFI-2; *Stober et al., 2016*). This break was crucial for minimizing potential aftereffects of TMS that could have affected performance during the post-phase and memory tasks (*Huang et al., 2005*; *Jannati et al., 2023*). After this break, participants engaged in a self-paced free recall task, after which they returned to the EEG room to complete the post-phase of the narrative-insight task. The comparison of neural activity patterns between the pre- and post-phases allowed the analysis of insight-related changes in neural memory representations. EEG recordings were obtained during the pre-, insight-, and post-phase of the narrative-insight task. Finally, participants completed a multi-arrangements task (*Kriegeskorte and Mur, 2012*) and a forced-choice recognition task. In total, the experiment took about 4.5 hr per participant and was completed within a single day.

## Narrative-insight task

To examine insight-related mnemonic integration processes, participants completed a modified version of the narrative-insight task (*Milivojevic et al., 2015*; *Figure 1*), while their brain activity was measured using EEG. The task involved watching life-like videos from the computer game *The Sims 3*, representing different storylines. Each storyline consisted of events that could either be integrated (A and B) into narratives or not (A and X). Participants were unaware that each narrative had two versions. The two versions shared event A but had different events B. Event X from one version served as event B in the other version. Thus, all participants viewed the same events A, B, and X, with 37 participants linking events A and X and 28 participants linking events A and B. Counterbalancing which events were linked over participants controlled for non-specific stimulus effects and visual similarity. Throughout this manuscript, the linked events are referred to as events A and B, while the non-linked event is referred to as X. Given the role of the angular gyrus in imagination (*Benoit and Schacter, 2015*; *Thakral et al., 2017*) and recent research highlighting distinct neural underpinnings when gaining insight via imagination vs. observation, we introduced two insight modes: imagination-based linking for half of the stories and observation-based linking for the other half.

The narrative-insight task consisted of three phases (*Figure 1*): the pre-phase, the insight-phase, and the post- phase. The task included 10 stories featuring three videos (A, B, and X) in both the pre- and post-phase, and two videos (L, C) and two imagination instructions (I, CI) in the insight-phase. In the *pre-phase*, participants viewed events A, B, and X for 2 s each, with inter-trial intervals (ITIs) between 700–1300ms (~1000 ms). Each video was preceded by a brief title (1 s) and presented 18 times in pseudorandom order. The presentation order ensured that each video appeared before the next round of presentations began, and consecutive trials did not feature the same video. Following the pre-phase, participants rated the extent to which they perceived the events as belonging together on a scale from not at all (1) to very much (4).

The subsequent *insight-phase* comprised two parts. For five stories, participants viewed the linking video event (L) interspersed with a control video event (C), each presented for 8 s and repeated nine times with ITIs between 700 and 1300ms (on average ~1000ms). For the other five stories, participants observed an instruction (I) to imagine a particular linking scene, with the video titles indicating the intended person, alternating with a control instruction (CI). Each instruction was presented for 8 s and repeated nine times with ITIs between 700 and 1300ms (~1000ms). A total of 32 participants first linked events via imagination and later via observation, while 33 participants first linked events via observation and then via imagination. Importantly, participants were stimulated with cTBS or sham before both parts of the insight-phase to maintain the stimulation effect throughout the insight-phase. For detailed description of the TMS procedure see below.

Participants were instructed to imagine specific scenes only when specifically asked to do so; otherwise, they were told to simply relax and watch the videos or answer the rating questions. In the observation condition, the linking video (L) depicted the main characters from videos A and B interacting with each other, while the control video (C) featured an unrelated activity involving an unknown character (e.g. two women engaged in conversation). In the imagination condition, a written linking instruction (I) prompted participants to imagine the main characters from video A and B interacting with each other, while the control instruction (CI) asked them to imagine an unknown character engaged in an unrelated activity (e.g. two women talking to each other). Following the insight-phase, participants provided ratings regarding their comprehension of the link and adherence to instructions on a scale ranging from 1 (not at all) to 4 (very much). After the insight-phase, participants were taken to another room for a 30-min break, during which they completed several questionnaires. After the break, they performed the free recall task (see below). Subsequently, in the final *post-phase*, events A, B, and X were again displayed for 2 s, repeated 18 times with ITIs between 700 and 1300ms (on average ~1000ms). Each video was preceded by a 1-s title. The post-phase order was pseudorandomized to minimize sequence effects. Its purpose was to examine neural representation changes for events A and B after participants learned that these were linked. Participants then rated the extent to which they perceived the events as belonging together on a scale of 1 (not at all) to 4 (very much). Participants received visual feedback in the form of highlighted selected responses when entering a rating question. In addition to the presentation of A, B, and X events in the pre- and post-phases, we included target events to ensure sustained attention throughout the experiment. These target events, accounting for 11% of pre- and post-phase

trials, required participants to press a button in response to a 2-s animated video of a girl on a pink scooter.

## Free recall

To assess the extent to which insight into the relationship of initially unrelated events affects subsequent memory, participants performed a free recall test in which they were instructed to recall all presented events in as much detail as possible (*Figure 1*). During free recall, participants were voice recorded for a maximum of 20 min. To assess the level of detailedness of the integrated episodes, audio recordings from free recall were scored according to how much detail of the different video events (A, B, and X) were recalled from day one and whether the linking events (L, and I) were named. A rating system was employed that allowed for distinct coding of details associated with each specific event (A, B, X, L, and I) and distinguished between central and peripheral details. Central details refer to elements that are crucial to the plot and directly impact the linking process. These details include significant aspects such as distinctive features of the protagonist in each event. Peripheral details encompass any observable details in the video events that are not central to the plot. For example, these could include features like the presence of a carpet in a room or the color of the curtains. Importantly, there was no difference in the number of details that could be named among the video events A, B, and X across different stories (*event: F*(1.35, 12.14) = 2.09, p = 0.173). However, it is worth noting that a greater number of peripheral details could be named compared to central details (*detail: F*(1, 9) = 83.24, *p<0.001*), which was expected as there were more details visible in the video events that were unrelated to the plot and, consequently, had no direct influence on the linking process. We engaged four independent raters and instructed them to assign details only to events for which it was clear that they belonged exclusively to that event, thereby avoiding any confusion between different events. The raters further scored whether the participants named the linking events or not. The scoring process involved two raters evaluating the first half of the data, while another two raters assessed the second half. All raters were blinded to the experimental conditions. To assess inter-rater reliability, all raters rated the first five participants, and on average, these ratings were highly correlated with each other (*mean correlation* = 0.80, SD = 0.14). To enhance inter-subjectivity, these ratings were averaged. The details of the different event types (A, B, and X) were combined across stories to generate a comprehensive rating of event details for both imagined and observed links. The average rating of the linked events (A and B) was then calculated to represent the overall measure of linked events. The non-linked event (X) remained unchanged. The naming of the linking events were combined across stories, separately for imagination and observation. Following the free recall, participants proceeded to the post-phase of the narrative-insight task.

## Multi-arrangements task

In order to ensure that participants accurately retained the structure of the events they gained insight into, we assessed their representational structure through a multi-arrangements task (*Kriegeskorte and Mur, 2012*). In this task, participants were instructed to arrange representative images (A, B, and X) from each story based on their relatedness. Using a computer mouse, participants dragged and dropped the images within a circular two-dimensional arena displayed on the computer screen. This task served to assess whether participants could successfully bring the linked images (A and B) closer together than the non-linked images (A and X). Each trial was self-paced and could be concluded by the participant by selecting 'Done'. In the first trial, participants had to arrange all images by similarity and were instructed to do so carefully. Subsequent trials consisted of subsets of the first trial selected based on an adaptive procedure designed to minimize uncertainty and better approximate the high-dimensional perceptual representational space. This procedure is based on an algorithm optimized to provide optimal evidence for the dissimilarity estimates (*Kriegeskorte and Mur, 2012*). Distances in this MA task were computed by initially computing the squared on-screen distance (Euclidian distance) between all items in the first trial to produce a roughly estimated representative dissimilarity matrix (RDM) and by iteratively updating this RDM by the weighted average of scaled trial estimates. The completion of the MA task required approximately 15 min. Distances for linked (A and B) and non-linked events (A and X) were averaged across stories for both imagined and observed links.

## Forced-choice recognition

To further ensure participants' accurate identification of linked and non-linked events following the narrative-insight task, a forced-choice recognition task was administered to assess participants' comprehension. They were presented with an image of event A at the top of the computer screen and had to indicate whether the image of B or X in the bottom half of the screen was linked to A. Participants were presented with these forced-choice options for each of the stories they had seen before. After indicating for a story which event was linked to event A, they had to rate how confident they were in their answer. Confidence was rated on a scale from not at all (1) to very sure (4). This process was repeated for all ten stories. Participants were presented with the forced-choice question and the confidence rating for 5.5 s each, with 1-s inter-stimulus intervals. Participants received visual feedback when submitting their ratings, as the selected response was highlighted. The forced-choice recognition test lasted approximately 5 min. Data from the forced-choice recognition task were pooled across stories and the percentage of correct responses was calculated, separately for imagined and observed links.

## Transcranial magnetic stimulation

Transcranial magnetic stimulation (TMS) was applied over the left angular gyrus before participants gained insight into the relationship of initially unrelated events. We used a PowerMAG Research 100 stimulator (MAG & More GmbH, München, Germany) for stimulation, that is specifically designed for delivering repetitive transcranial magnetic stimulation (rTMS) in both clinical and research applications. Two identically looking but different 70 mm figure-of-eight-shaped coils were used depending on the TMS condition: The PMD70-pCool coil (MAG & More GmbH, München, Germany) with a 2T maximum field strength was used for cTBS, while the PMD70-pCool-SHAM coil (MAG & More GmbH, München, Germany), with minimal magnetic field strength, was employed for sham, providing a similar sensory experience, with stimulation pulses being scattered over the scalp and not penetrating the skull.

## Motor threshold determination

The motor threshold (MT) was assessed at the beginning of the experiment while participants were at rest, wearing an EEG cap without electrodes attached. This measurement was utilized to determine the appropriate strength of TMS required to pass through the cap. Disposable Ag/AgCL surface electromyography (EMG) electrodes were placed on the right abductor pollicis brevis (ABP) muscle, with the reference electrode on the bony landmark of the index finger and the ground electrode on the right elbow. To locate the motor hotspot, we identified the center of the head and moved 5 cm to the left and 3.5 cm forward at a 45° angle, marking it as the center point of a 9-point grid search area with each point spaced 1 cm apart from adjacent points. Starting at 40% of the maximum stimulator output (MSO), we gradually increased the intensity in 5% increments while positioning the TMS coil at a 45° angle and moving it around the search area, delivering single pulses until we identified the motor hotspot. Once the motor hotspot was located, the MT was determined at that site. It was defined as the minimum percentage of maximum stimulator output (MSO) over the left motor cortex needed to elicit motor evoked potentials (MEPs) with a peak-to-peak amplitude of 50 µV in eight out of 16 consecutive pulses.

## Neuro-navigation

Before the experimental session, we obtained individual T1-weighted structural MR images using a 3T Siemens PRISMA scanner from each participant. These images were used for neuro-navigation with the PowerMag View! System (MAG & More GmbH, München, Germany). The system utilizes two infrared cameras (Polaris Spectra) to track the positions of the participant's head and TMS coil in space. Based on the individual T1 MR images, we created 3D reconstructions of the participants' heads, allowing us to precisely locate the left angular gyrus coordinate (MNI: −48,−67, 30), initially derived from previous work (MNI: −48,−64, 30; *Thakral et al., 2017*), for TMS stimulation. Despite a minor deviation in coordinates due to necessary MNI to Talairach transformations for software compatibility (Powermag View! by MAG & More GmbH, München, Germany), our methodology ensured precise localization of the angular gyrus target area. The coordinates were entered as TAL coordinates. Once the TAL coordinate was entered, the coil was positioned in accordance with the template provided by the neuro-navigation system, aiming for a brain-to-target distance of less than 3 cm. This procedure

ensured precise coil placement tailored to the unique anatomy of each participant, while maintaining the shortest and therefore optimal distance to the cortex.

## Continuous theta burst stimulation

Depending on the experimental condition, we administered continuous Theta Burst Stimulation (cTBS) using either the cTBS or the sham figure-of-eight coil at 80% of the MT intensity. The experiment was conducted in a double-blind manner, where neither the participant nor the experimenter were aware of the stimulation condition (cTBS vs. sham). Previous evidence has demonstrated the inhibitory effect of cTBS on the targeted brain region under stimulation (*Huang et al., 2005*; *Jannati et al., 2023*). Nonetheless, the effects of cTBS appear to vary based on the targeted region, with cTBS to parietal regions demonstrating the capability to enhance hippocampal connectivity (*Hermiller et al., 2019*; *Hermiller et al., 2020*). Following the standard cTBS protocol, participants received a series of bursts comprising three magnetic pulses (pulse triplets) at a frequency of 50 Hz, with the triplets repeated at a rate of five Hz (i.e. five pulse triplets per second). Each participant received a total of 600 magnetic pulses delivered over a 40-s duration. The coil was positioned tangentially on the head and mechanically fixed in a coil holder, with its handle pointing upwards to maintain its position. Throughout the stimulation, it was ensured via neuro-navigation that the brain-to-target distance remained below 3 cm from the left angular gyrus coordinate (MNI: −48,−67, 30).

## Electric field modeling

Electric field simulations were performed in SimNIBS v4.0.1 to perform TMS simulations for the cTBS group. To assess the potential stimulation strength based on individual motor thresholds and T1 images, we segmented and meshed these MRI scans into tetrahedral head models using the SimNIBS charm pipeline. All head models were visually inspected to exclude segmentation errors. In a next step, we performed the TMS simulation at 80% of individual motor thresholds ($M$ = 54.82%; SD = 12.93 %). We modeled the Magstim 70 mm figure-of-eight coil placed over the left angular gyrus target coordinates (MNI: −48,−67, 30), accounting for the presence of the EEG cap during stimulation. Next, to estimate the average field strength in the region of interest (ROI), we extracted the gray matter regions and created a 10 mm spherical ROI centered around the target coordinate and averaged the estimated field strength for the sphere. This approach enabled us to evaluate the potential stimulation strength and its impact on the target brain area (*Pizem et al., 2022*; *Zhang et al., 2022*).

## EEG data acquisition

EEG data was recorded using a 64-electrode BioSemi ActiveTwo system (BioSemi B.V., Amsterdam, the Netherlands) following the international 10–20 system. The sampling rate was set to 1024 Hz, and a band-pass filter of 0.03–100 Hz was applied online. Additional electrodes were placed at the mastoids, above and below the orbital ridge of the right eye, and at the outer canthi of both eyes. We maintained impedances within a range of ±20 μV using the common mode sense (CMS) and driven right leg (DRL) electrodes, serving as active reference and ground, respectively.

## Behavioral data analysis

For our behavioral analyses we opted to employ linear-mixed models (LMM), given their high robustness regarding the underlying distribution and high sensitivity to individual variation (*Pinheiro and Bates, 2000*; *Schielzeth et al., 2020*). To illuminate the impact of gaining insight into the relationship between initially unrelated events on subsequent memory, we subjected the number of details remembered during free recall to a LMM implemented with the lme4 package (*Bates et al., 2015*) including group (cTBS/sham), mode (imagination/observation), link (link/non-link), and detail (central/peripheral) and their interactions as fixed effects, with a random intercept per participant. As a follow-up analysis, we calculated a median split on the cTBS group, based on simulations of the electric field strength and re-analyzed this data with a new group variable (sham/low stim/high stim), mode (imagination/observation), link (link/non-link), and detail (central/peripheral) and their interactions as fixed effects, with a random intercept per participant. To further examine mnemonic integration based on memory of the linking phase, naming of the linking events were entered into an LMM including group (cTBS/sham), and mode (imagination/observation), and their interactions as fixed effects. Additionally, a random intercept per participant was included to account for individual variability. As a follow-up

analysis, we re-analyzed this data with a new group variable (sham/low stim/high stim), mode (imagination/observation), link (link/non-link), and detail (central/peripheral) and their interactions as fixed effects, with a random intercept per participant to estimate the effect of stimulation strength on this outcome. To verify that all participants acquired insight into the relationship between events, we analyzed the ratings for the event duplets of interest (linked events AB and non-linked events AX) from the pre- and post-phase of the narrative-insight task. This analysis was conducted using a LMM including group (cTBS/sham), mode (imagination/observation), time (pre/post), and link (link/non-link) and their interactions as fixed effects and a random intercept per participant. To confirm the retention of the representational structure of the narrative-insight task in memory, Euclidean distance estimates were extracted from the multi-arrangements task for linked (AB) and non-linked events (AX), averaged across stories, and then entered into an LMM including group (cTBS/sham), mode (imagination/observation), and link (link/non-link), and their interactions as fixed effects and a random intercept per participant. Additionally, to further ensure participants' accurate identification of linked and non-linked events, we assessed performance in the forced-choice recognition task by calculating the proportion of correct answers. These performance measures (in %) were then entered into an LMM including group (cTBS/sham), and mode (imagination/observation), and their interactions as fixed effects and a random intercept per participant.

All analyses were performed in R version 4.0.4 and for all analyses standardized betas are reported. Prior to the analysis, the data were examined for outliers, defined as mean +/-3 *SD*. For the modified narrative-insight task analysis, three outliers (two from the sham and one from the cTBS group) were identified and excluded. For the analysis of the forced-choice recognition task, two outliers (one from each group) were identified and excluded. For the analysis of the multi-arrangements task, four outliers (one from the sham and three from the cTBS group) were identified and excluded. For the free recall analysis, five outliers (two from the sham and three from the cTBS group) were identified and excluded. After outlier correction, we identified non-normality in our data using a Shapiro-Wilk test (narrative-insight task: $W = 0.92$, $p<0.001$; multi-arrangements task: $W = 0.94$, $p<0.001$; forced-choice recognition: $W = 0.50$, $p<0.001$; free recall details: $W = 0.85$, $p<0.001$; free recall naming of linking events: $W = 0.94$, $p<0.001$). However, we mitigated this by employing linear-mixed models (LMMs), recognized for their robustness even with non-normally distributed data (*Schielzeth et al., 2020*).

## EEG preprocessing

The offline analysis of EEG data from the narrative-insight task was conducted using the FieldTrip toolbox (*Oostenveld et al., 2011*) and custom scripts implemented in Matlab (TheMathWorks). Pre- and post-phase trials were segmented from –2 to 3 relative to stimulus onset and then re-referenced to the mean average of all scalp electrodes. The data were demeaned based on the average signal of the entire trial and de-trended. To eliminate power-line noise, a discrete Fourier-Transform filter (DFT) at 50 Hz was applied. Any electrodes that did not record or exhibited constant noise were removed (max. one per participant) and interpolated using weighted neighboring electrodes. Noisy trials were removed after visual inspection, on average 2.32 (+/-*SD* 1.34) of the 540 pre-phase trials and 2.63 (+/-*SD* 1.74) of the 540 post-phase trials. Following artifact rejection, the epochs were down-sampled to 256 Hz. Next, we performed an extended infomax independent component analysis (ICA) using the 'runica' method with a stop criterion of weight change $<10^{-7}$ to identify and reject components associated with eye blinks and other sources of noise. In a two-step procedure, we first correlated the signals from the horizontal and vertical EOG electrodes with each independent component. Components exhibiting a correlation higher than 0.9 were immediately removed from further analysis. In a second step, the remaining components were identified through visual inspection of their time courses and corresponding brain topographies. On average, 3.32 (+/-*SD* 1.38) components were removed before back projecting the signals into electrode space.

## Representational similarity analysis

To investigate how the brain processes insight-induced changes in the relationships between unrelated events, we conducted a representational similarity analysis (RSA) at the EEG electrode level (*Heinbockel et al., 2022*; *Pacheco Estefan et al., 2021*). This method is ideally suited to measure neural representation changes and was specifically chosen as it has been previously identified as the preferred approach for quantifying insight-induced neural changes (*Grob et al., 2023b*;

*Milivojevic et al., 2015*). RSA allows us to estimate neural activity patterns associated with specific events by measuring their correlations, thus providing insights into the underlying neural processes (*Kriegeskorte et al., 2008*). To measure the insight-induced representational changes, we focused on assessing the similarity of linked and non-linked events before and after gaining insight, separately for events that were linked via imagination and events that were linked via observation. We performed the RSA in the theta frequency range as prior evidence highlighted the key role of theta activity in episodic memory integration (*Backus et al., 2016*; *Nicolás et al., 2021*).

To perform this analysis, we first calculated time-frequency representations utilizing spectral decomposition using sliding Hanning windows on the preprocessed EEG data. The frequency range was set from 2 Hz to 45 Hz, with 1 Hz increments and a five-cycle window. The analysis was conducted within a time interval of –2–3 s relative to stimulus onset. For each participant, single trial power estimates were then averaged across stories and baseline corrected using absolute baseline correction with a time window of –1.8 to –1 s relative to stimulus onset. The time-frequency data was then appended into separate data files for the pre- and post-phase, as well as for imagined and observed stories. In a second step, we utilized the time-frequency data obtained in the theta range (4–7 Hz) to conduct RSA. These theta power values were then combined to create representational feature vectors, which consisted of the power values for four frequencies (4–7 Hz) × 41 time points (0–2 s) × 64 electrodes. We then calculated Pearson's correlations to compare the power patterns across theta frequency between the time points of linked events (A with B), as well as between the time points of non-linked events (A with X) for the pre- and the post-phase separately, separately for stories linked via imagination and via observation. To ensure unbiased results, we took precautions not to correlate the same combination of stories twice, which prevented potential inflation of the data. To facilitate statistical comparisons, we applied a Fisher z-transform to the Pearson's *rho* values at each time point. This yielded a global measure of similarity on each electrode site. We, thus, obtained time × time similarity maps for the linked events (A and B) and the non-linked events (A and X) in the pre- and post-phases, separately for insight gained through imagination and observation. In total, this analysis produced eight Representational Dissimilarity Matrices (RDMs) for each electrode and each participant.

We performed statistical analyses on the RDMs using cluster-based permutation t-tests in the Fieldtrip toolbox (10.000 permutations; *Oostenveld et al., 2011*). This approach allows for testing statistical differences while controlling for multiple comparisons without spatial constraints (*Maris and Oostenveld, 2007*). The samples were clustered at a level of $\alpha_{cluster}$ = 0.001 to allow for more refined clusters. Clusters with a corrected Monte Carlo *p-value* <0.05 were considered statistically significant. The RDMs for the change from pre- to post-phase in linked events (post - pre) that were linked via imagination (vs. observation; imagination - observation) were contrasted between the cTBS (vs. the sham) groups via an independent sample cluster-based permutation t-test. Similarly, the RDMs for the change from the pre- to post-phase (post - pre) in non-linked events that were linked via imagination (vs. observation) were contrasted between the cTBS (vs. sham) groups via an independent sample cluster-based permutation t-test.

## Coherence analysis

Due to the robust connectivity between the angular gyrus and other brain regions (*Petit et al., 2023*; *Seghier, 2013*), we proceeded with a connectivity analysis as a next step. To gain a deeper understanding of the connectivity changes between linked (A and B) events before and after gaining insight, we conducted a sliding window coherence analysis in electrode-space on the same contrasts that we found in our RSA analyses. Therefore, this analysis specifically focused on the comparison between imagined and observed links, utilizing the concept of imaginary coherence (*Nolte et al., 2004*). Imaginary coherence quantifies the synchronization between two electrodes, accounting for phase-lag at a specific frequency and minimizing the influence of volume conduction effects. We first computed a frequency analysis focusing on the mean theta frequency (5.5 Hz; dpss-taper = 1.5 Hz). Then, we computed imaginary coherence for all possible electrode combinations. To capture the temporal dynamics, we employed a sliding window approach that spanned the duration of the video display (0–2 s) in 500ms windows, sliding forward in steps of 50ms. By applying this sliding window analysis, we obtained a coherence spectrum matrix of 64 (electrodes) × 64 (electrodes) for each of the 31 time windows in each participant. We calculated the average coherence matrices for both A and B, resulting in a single coherence matrix that represents the connectivity patterns of the linked events.

Subsequently, we focused on the same interaction that was yielded by the previous RSA by examining the changes in coherence spectra from pre- to post-phase (post - pre) for the imagined (vs. observed) linked events. To determine the statistical significance of the observed differences between groups (sham vs. cTBS), we utilized an independent sample cluster-based permutation t-test across all 31 time windows, correcting for the multiple comparisons of channels and time windows.

In order to investigate the changes in connectivity from pre- to post-phase for the non-linked event (X), we conducted a sliding window coherence analysis following the same procedure as described above. Please note that the non-linked event was treated separately and not averaged with any other event to maintain its distinct characteristics. Similar to the linked events analysis, we obtained 31 time windows representing the change from pre- to post-phase (post - pre) for the imagined (vs. observed) non-linked event, which was the same interaction as in the previous RSA. To evaluate the statistical significance of these changes between the groups (sham vs. cTBS), we employed an independent sample cluster-based permutation t-test.

## Correlational analysis

To relate the findings from the RSA and the coherence analyses to the behavioral results, we extracted the significant clusters. We then proceeded to estimate the correlation between the extracted neural cluster activity and behavioral outcomes. Specifically, we correlated neural similarity and activity with behavioral outcomes separately for the cTBS and sham groups. Subsequently, we compared these correlations to determine if they differed significantly from each other (*Eid et al., 2017*).

## Control variables

In order to ensure that the observed effects were solely attributable to the TMS manipulation and not influenced by other factors, we comprehensively evaluated several trait and state variables. To account for potential variations in anxiety levels that could impact our results, we specifically measured state and trait anxiety using STAI-S and STAI-T (*Laux et al., 1981*), thus minimizing the potential confounding effects of anxiety on our findings (*Charpentier et al., 2021*). Additionally, we evaluated participants' chronic stress levels using the TICS (*Schulz and Schlotz, 1999*) to exclude any group variations that might explain the effect on memory, cosidering the well-established impact of stress on memory (*Sandi and Pinelo-Nava, 2007*; *Schwabe et al., 2012*). Moreover, we assessed participants' depressive symptoms employing the BDI (*Hautzinger et al., 2006*), to guarantee group comparability on this clinical measure. We further assessed fundamental personality dimensions using the BFI-2 (*Stober et al., 2016*) to exclude any potential group discrepancies that could account for differences observed. Lastly, we assessed participants' imaginative capacities using the FFIS (*Zabelina and Condon, 2020*), to ensure uniformity across groups regarding this central variable, considering the significant role of imagination in relation to the cTBS-targeted angular gyrus (*Thakral et al., 2017*).

## Acknowledgements

Funding This project was funded by the German Research Foundation (DFG; grant SCHW1357/22 to LS and CFD). CFD's research is supported by the Max Planck Society, the European Research Council (ERC-CoG GEOCOG 724836), the Kavli Foundation, the Jebsen Foundation, Helse Midt Norge, and the Research Council of Norway (223262 /F50 and 197467 /F50).

## Additional information

### Funding

| Funder | Grant reference number | Author |
| --- | --- | --- |
| Deutsche Forschungsgemeinschaft | SCHW1357/22 | Christian F Doeller Lars Schwabe |
| European Research Council | ERC-CoG GEOCOG 724836 | Christian F Doeller |
| Kavli Foundation | 223262/F50 | Christian F Doeller |

| Funder | Grant reference number | Author |
| --- | --- | --- |
| Helse Midt-Norge | 197467/F50 | Christian F Doeller |

The funders had no role in study design, data collection and interpretation, or the decision to submit the work for publication.

## Author contributions

Anna-Maria Grob, Conceptualization, Data curation, Formal analysis, Investigation, Visualization, Methodology, Writing - original draft, Writing - review and editing; Hendrik Heinbockel, Formal analysis, Visualization, Methodology, Writing - review and editing; Branka Milivojevic, Christian F Doeller, Conceptualization, Writing - review and editing; Lars Schwabe, Conceptualization, Resources, Supervision, Funding acquisition, Writing - original draft, Writing - review and editing

## Author ORCIDs

Anna-Maria Grob  http://orcid.org/0000-0001-6108-6432
Christian F Doeller  http://orcid.org/0000-0003-4120-4600
Lars Schwabe  http://orcid.org/0000-0003-4429-4373

## Ethics

All participants gave written informed consent before participation and received a monetary compensation at the end of the experiment. The procedures were approved by the local ethics committee (Faculty of Psychology and Human Movement Science, Universität Hamburg, Hamburg, Germany, 2020_301 Grob Schwabe) and adhered to the Declaration of Helsinki.

Reviewer #2 (Public Review): https://doi.org/10.7554/eLife.91033.3.sa1
Reviewer #3 (Public Review): https://doi.org/10.7554/eLife.91033.3.sa2
Author Response https://doi.org/10.7554/eLife.91033.3.sa3

# Additional files

## Supplementary files

- MDAR checklist
- Supplementary file 1. Supplementary material.

## Data availability

The data generated in this study as well as all original code is available at https://doi.org/10.25592/uhhfdm.12928. For any additional information needed to reanalyze the reported data, please contact the lead researcher directly.

The following dataset was generated:

| Author(s) | Year | Dataset title | Dataset URL | Database and Identifier |
| --- | --- | --- | --- | --- |
| Grob A-M, Heinbockel H, Milivojevic B, Doeller C, Schwabe L | 2023 | Causal role of the angular gyrus in insight-driven memory reconfiguration | https://doi.org/10.25592/uhhfdm.12928 | ZFDM Repository - Universität Hamburg, 10.25592/uhhfdm.12928 |

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
