## [Editor Report · eLife assessment]

This **important** paper provides **solid** evidence that the angular gyrus plays a role in insight-based memory updating. The study is well conducted, timely, and presents clear-cut behavioral results. While the study provides robust evidence that transcranial magnetic stimulation to the angular gyrus impacts memory, evidence for the strong claim of a causal contribution of the angular gyrus in particular – apart from other connected regions, including the hippocampus – is not conclusive.

---

## [Referee Report · Reviewer #2 (Public Review)]

The formation of long-term memory representations requires the continuous updating of ongoing representations. Various studies have shown that the left angular gyrus (AG) may support this cognitive operation. However, this study demonstrates that this brain region plays a causal role in the formation of long-term memory representations, affecting both the neural and behavioural measures of information binding.

A significant strength of this work is that it is the first one to test the hypothesis that the left angular gyrus has a causal role in the reconfiguration and binding of long-term memory representations by comparing when insights are primarily derived from direct observation versus imagination. Consequently, the results from this manuscript have the potential to be informative for all areas of cognitive research, including basic perception, language cognition and memory.

Furthermore, this study presents a comprehensive set of measurements on the same individuals, encompassing various task-related behavioural measures, EEG data, and questionnaire responses.

A weakness of the manuscript is the use of different groups of participants for the key TMS intervention.

---

## [Referee Report · Reviewer #3 (Public Review)]

The authors have done a fine job of updating the manuscript and it is substantially improved. In particular, the paragraphs towards the end of the Introduction and Discussion are vastly improved. The last paragraph of the Introduction now clearly explicates the hypotheses (save one minor point of confusion). The limitations section of the Discussion is also very helpful and fair. However, there are still areas where claims need to be tempered.

Major criticisms

• The results still do not lead to the conclusion that the angular gyrus is causally involved in insight-driven memory configuration. Although the authors do state that other regions such as the hippocampus may have contributed to the pattern of results, there is still no evidence of target engagement or a link between target engagement and the behavioral results. Thus, while the results support that cTBS to the angular gyrus affects insight-driven memory configuration, it is a strong overstep to say that the angular gyrus is causally involved in insight-driven memory reconfiguration. In particular, this applies to both the title and the last line of the Abstract. In relation to this, have the authors conducted any target engagement analyses? It seems like a good starting point would be to identify the censor closest to the stimulation site in each individual, Hjorth transforms the signal of that sensor by subtracting the average of the surrounding sensors to increase signal localization, and then measure the effects of stimulation on theta power. Presumably, we would expect that cTBS would decrease theta power relative to sham stimulation. Although this isn't the only type of analysis that could at least partially confirm target engagement, there needs to be some sort of formal analysis to maintain the claims of the title and last line of the Abstract.

• The authors removed the mentions of "inhibitory stimulation" from the manuscript to their credit, but a rigorous and fair treatment of the effects of cTBS is still lacking, and it is still unclear why cTBS to the angular gyrus would cause an inhibitory effect in the first place. The authors state that

"Previous evidence has demonstrated the inhibitory effect of cTBS on the targeted brain region under stimulation (Huang et al., 2005; Jannati et al., 2023). Nonetheless, the effects of cTBS appear to vary based on the targeted region, with cTBS to parietal regions demonstrating the capability to enhance hippocampal connectivity (Hermiller et al., 2019, 2020)."

The inhibitory effects of motor cortex cTBS s on corticospinal excitability in nine subjects from the Huang paper and the Jannati review (not a primary source) do not constitute sufficient evidence to hypothesize an inhibitory effect on insight-driven memory reconfiguration. The second sentence provides much more sufficient evidence that parietal stimulation should have some sort of a facilitatory effect, but this is simply glossed over without an explanation of why cTBS to the parietal cortex should inhibit insight-driven memory reconfiguration. Pilot data showing such inhibitory effects or a body of evidence showing inhibitory effects of angular gyrus stimulation on closely-related areas of cognition would have given reason to believe this. However, without these, an a priori assumption that parietal cTBS would be inhibitory seems highly debatable and paints the results as provisional, rather than confirmatory"

---

## [Author Response]

Responses to public reviews

**Reviewer 1**
We thank the reviewer for the valuable and constructive comments and are pleased that the reviewer finds our study timely and our behavioral results clear.1. The RSA basically asks on the lowest level, whether neural activation patterns (as measured by EEG) are more similar between linked events compared to non-linked events. At least this is the first question that should be asked. However, on page 11 the authors state: "We ex-amined insight-induced effects on neural representations for linked events [...]". Hence, the critical analysis reported in the manuscript fully ignores the non-linked events and their neu-ral activation patterns. However, the non-linked events are a critical control. If the reported effects do not differ between linked and non-linked events, there is no way to claim that the effects are due to experimental manipulation - neither imagination nor observation. Hence, instead of immediately reporting on group differences (sham vs. control) in a two-way in-teraction (pre vs. post X imagination vs. observation), the authors should check (and re-port) first, whether the critical experimental manipulation had any effect on the similarity of neural activation patterns in the first place.

We completely agree that the non-link items are a critical control. Therefore, we had reported not only the results for linked but also for non-linked events on page 15, lines 336-350. We clarified this important point now on page 12 lines 283-286:

“Subsequently, we examined insight-induced effects on neural representations for linked (vs. non-linked) events by comparing the change from pre- to post-insight (post-pre) and the difference between imagination and observation (imagination - observation) between cTBS and sham groups using an independent cluster-based permutation t-test.”

Moreover, to directly compare linked and non-linked events we performed a four-way in-teraction including link vs. non-link. This analysis yielded a significant four-way interaction, showing that the interaction of time (pre vs. post), mode of insight (imagination vs. obser-vation) and cTBS differed for linked vs. non-linked items. We then report the follow-up analyses, separately for linked and non-linked events. Please see pages 12-13, lines 287-294:

“First, we included the within-subject factors time (pre vs. post), mode of insight (imagina-tion vs. observation) and link (vs. non-link) by calculating the difference waves. Subse-quently we conducted a cluster-based permutation test comparing the cTBS and the sham groups. This analysis yielded a four-way interaction within a negative cluster in a fronto-temporal region (electrode: FT7; p = 0.007, ci-range = 0.00, SD = 0.00). This result indicates that the impact of cTBS over the angular gyrus on the neural pattern reconfiguration follow-ing imagination- vs. observation-based insight may differ between linked and non-linked events. For linked events, this analysis yielded a […]”

1. Overall, the focus on the targeted three-way interaction is poorly motivated. Also, a func-tional interpretation is largely missing.

In order to better explain our motivation for the three-way interaction, we em-phasized in the introduction the importance of disentangling potential differences due to the mode of insight, given the known role of the angular gyrus in imagination on pages 4-5, lines 107-115:

“Considering this involvement of the angular gyrus in imaginative processes, we expected that the effect of cTBS on the change in representational similarity from pre- to post-insight will differ based on the mode of insight – whether this insight was gained via imagination or observation. Specifically, we expected a more pronounced impairment in the neural recon-figurations when insight is gained via imagination, as this function may depend more on an-gular gyrus recruitment than insight gained via observation. Additionally, we expected cTBS to the left angular gyrus to interfere with the increase in neural similarity for linked events and with the decrease of neural similarity for non-linked event.”

As discussed on page 21 (starting from line 478; see also the intro on page 4), we expected that the angular gyrus would be particularly implicated in imagination-based insight, given its known role in imagination (e.g.: Thakral et al., 2017). Moreover, given the angular gyrus’s strong connectivity with other regions, the results observed may not be driven by this re-gion alone but also by interconnected regions, such as the hippocampus. We clarified these important points at the very end of the discussion on pages 23-24, lines 543-560:

“Furthermore, the differential impact of cTBS to the angular gyrus on neural reconfigura-tions between events linked via imagination and those linked via observation may be at-tributed to its crucial role in imaginative processes (Ramanan et al., 2018; Thakral et al., 2017). Another intriguing aspect to consider is that the stimulated site was situated in the more ventral portion of the angular gyrus, recognized for its stronger connectivity to the episodic hippocampal memory system in contrast to its more dorsal counterpart (Seghier, 2013; Uddin et al., 2010). This stronger connectivity between the ventral angular gyrus and the hippocampus may shed light on the greater impact of cTBS to the angular gyrus on im-agination-based insight. Given the angular gyrus’s robust connectivity with other brain re-gions, including the hippocampus (Seghier, 2013), it is plausible that the observed changes might not solely stem from alterations within the angular gyrus itself, but could also origi-nate from these interconnected regions. This notion may bear particular importance given the required accessibility to the hippocampus during imaginative processes (Benoit & Schacter, 2015; Grob et al., 2023a; Zeidman & Maguire, 2016). Interactions between the an-gular gyrus and the hippocampus may give rise to rich memory representations (Ramanan et al., 2018). In line with this, recent studies have demonstrated that cTBS to the angular gy-rus resulted in enhanced hippocampal connectivity and improved associative memory (Hermiller et al., 2019; Tambini et al., 2018; Wang et al., 2014).”

1. "Interestingly, we observed a different pattern of insight-related representational pattern changes for non-linked events." It is not sufficient to demonstrate that a given effect is pre-sent in one condition (linked events) but not the other (non-linked events). To claim that there are actually different patterns, the authors would need to compare the critical condi-tions directly (Nieuwenhuis et al., 2011).

We completely agree and now compared the two conditions directly. Specifical-ly, we now report the significant four-way interaction, including the factor link vs. non-link, before delving into separate analyses for linked and non-linked events on pages 12-13, lines 287-294:

“First, we included the within-subject factors time (pre vs. post), mode of insight (imagina-tion vs. observation) and link (vs. non-link) by calculating the difference waves. Subse-quently we conducted a cluster-based permutation test comparing the cTBS and the sham groups. This analysis yielded a four-way interaction within a negative cluster in a fronto-temporal region (electrode: FT7; p = 0.007, ci-range = 0.00, SD = 0.00). This result indicates that the impact of cTBS over the angular gyrus on the neural pattern reconfiguration follow-ing imagination- vs. observation-based insight may differ between linked and non-linked events. For linked events, this analysis yielded a […]”

1. "This analysis yielded a negative cluster (p = 0.032, ci-range = 0.00, SD = 0.00) in the parieto-temporal region (electrodes: T7, Tp7, P7; Fig. 3B)." (p. 11). The authors report results with specificity for certain topographical locations. However, this is in stark contrast to the fact that the authors derived time X time RSA maps.

We did derive time × time similarity maps for each electrode within each partic-ipant, which allowed us to find a cluster consisting of specific electrodes. We apologize for not making this aspect clear enough and have, therefore, modified the respective part of our methods section on page 38, lines 951-952:

“In total, this analysis produced eight Representational Dissimilarity Matrices (RDMs) for each electrode and each participant.”

1. "These theta power values were then combined to create representational feature vectors, which consisted of the power values for four frequencies (4-7 Hz) × 41 time points (0-2 sec-onds) × 64 electrodes. We then calculated Pearson's correlations to compare the power pat-terns across theta frequency between the time points of linked events (A with B), as well as between the time points of non-linked events (A with X) for the pre- and the post-phase separately, separately for stories linked via imagination and via observation. To ensure un-biased results, we took precautions not to correlate the same combination of stories twice, which prevented potential inflation of the data. To facilitate statistical comparisons, we ap-plied a Fisher z-transform to the Pearson's rho values at each time point. This yielded a global measure of similarity on each electrode site. We, thus, obtained time × time similarity maps for the linked events (A and B) and the non-linked events (A and X) in the pre- and post-phases, separately for the insight gained through imagination and observation." (p. 34+35).If RSA values were calculated at each time point and electrode, the Pearson correlations would have been computed effectively between four samples only, which is by far not enough to derive reliable estimates (Schönbrodt & Perugini, 2013). The problem is aggra-vated by the fact that due to the time and frequency smoothing inherent in the time-frequency decomposition of the EEG data, nearby power values across neighboring theta frequencies are highly similar to start with. (e.g., Schönauer et al., 2017; Sommer et al., 2022).Alternative approaches would be to run the correlations across time for each electrode (re-sulting in the elimination of the time dimension) or to run the correlations at each time point across electrodes (resulting in the elimination of topographic specificity).At least, the authors should show raw RSA maps for linked and non-linked events in the pre- and post-phases separately for the insight gained through imagination and observa-tion in each group, to allow for assessing the suitability of the input data (in the supple-ments?) before progressing to reporting the results of three-way interactions.

Although we do see the reviewer’s point, we think that an RSA specific to the theta range yielding electrode specific time × time similarity maps must be run this way, otherwise, as you pointed out, one or the other dimension is compromised. Running an RSA across time for each electrode will lead to computing a similarity measure between the events without information on when these stimuli become more or less similar, thereby ig-noring the temporal dynamics crucial to EEG data and not taking advantage of the high temporal resolution. Conversely, conducting an RSA across electrodes might result in an overall similarity measure per participant, disregarding the spatial distribution and potential variations among electrodes. Although EEG has limited spatial resolution, different elec-trodes can capture differences that may aid in understanding neural processing. However, as suggested by the reviewer, we included the raw RSA maps for linked and non-linked events separately for pre- and post-phases, imagination and observation and link and non-link in the supplement and refer to these data in the results section on pages 12-13, lines 293-295:

“For linked events, this analysis yielded a negative cluster (p = 0.032, ci-range = 0.00, SD = 0.00) in the parieto-temporal region (electrodes: T7, Tp7, P7; Fig. 3B; Figure 3 – Figure sup-plement 1).”

And on page 15, lines 339-341:

“This analysis yielded a positive cluster (p = 0.035, ci-range = 0.00, SD = 0.00) in a fronto-temporal region (electrode: FT7; Fig. 3C; Figure 3 – Figure supplement 2).”

**Reviewer 2**

We thank the reviewer for the very helpful and constructive comments and appreciate that the reviewer finds our study relevant to all areas of cognitive research.

1. While the observed memory reconfiguration/changes are attributed to the angular gyrus in this study, it remains unclear whether these effects are solely a result of the AG's role in re-configuration processes or to what extent the hippocampus might also mediate these memory effects (e.g., Tambini et al., 2018; Hermiller et al., 2019).

We agree that, in addition to the critical role of the angular gyrus, there may be an involvement of the hippocampus. We point now explicitly to the modulatory capacities of angular gyrus stimulation on the hippocampus. Please see page 4, lines 81-88:

“One promising candidate that may contribute to insight-driven memory reconfiguration is the angular gyrus. The angular gyrus has extensive structural and functional connections to many other brain regions (Petit et al., 2023), including the hippocampus (Coughlan et al., 2023; Uddin et al., 2010). Accordingly, previous studies have shown that stimulation of the angular gyrus resulted in altered hippocampal activity (Thakral et al., 2020; Wang et al., 2014). Furthermore, the angular gyrus has been implicated in a myriad of cognitive func-tions, including mental arithmetic, visuospatial processing, inhibitory control, and theory-of-mind (Cattaneo et al., 2009; Grabner et al., 2009; Lewis et al., 2019; Schurz et al., 2014).”

We further added a new paragraph to the discussion pointing at the possibility that not solely the angular gyrus but another brain region, such as the hippocampus, may have me-diated the changes observed in our study on pages 23-24, lines 546-562:

“Another intriguing aspect to consider is that the stimulated site was situated in the more ventral portion of the angular gyrus, recognized for its stronger connectivity to the episodic hippocampal memory system in contrast to its more dorsal counterpart (Seghier, 2013; Ud-din et al., 2010). This stronger connectivity between the ventral angular gyrus and the hip-pocampus may shed light on the greater impact of cTBS to the angular gyrus on imagination-based insight. Given the angular gyrus’s robust connectivity with other brain regions, includ-ing the hippocampus (Seghier, 2013), it is plausible that the observed changes might not solely stem from alterations within the angular gyrus itself, but could also originate from these interconnected regions. This notion may bear particular importance given the re-quired accessibility to the hippocampus during imaginative processes (Benoit & Schacter, 2015; Grob et al., 2023a; Zeidman & Maguire, 2016). Interactions between the angular gyrus and the hippocampus may give rise to rich memory representations (Ramanan et al., 2018). In line with this, recent studies have demonstrated that cTBS to the angular gyrus resulted in enhanced hippocampal connectivity and improved associative memory (Hermiller et al., 2019; Tambini et al., 2018; Wang et al., 2014). However, it should be noted that our study detected impaired associative memory following cTBS to the angular gyrus.”

1. Another weakness in this manuscript is the use of different groups of participants for the key TMS intervention, along with underspecified or incomplete hypotheses/predictions.

In our view, the chosen between-subjects design is to be preferred over a crossover design for several reasons. First, our choice aimed to eliminate potential se-quence effects that may have adversely affected performance in the narrative-insight task (NIT). Second, this approach ensured consistency in expectations regarding the story links while also mitigating potential differences induced by fatigue. Additionally, we accounted for the potential advantage of a within-subject design – the stimulation of the same brain – by utilizing neuro-navigated TMS for targeting the stimulation coordinate. Finally, it is im-portant to note that we measured the event representations pre- and post-insight and that also the mode of insight was manipulated within-subject. Thus, our design did include a within-subject component and we are convinced that the chosen paradigm balances the different strengths and weaknesses of within-subject and between-subjects designs in the best possible manner. We specified our rationale for choosing a between-subjects ap-proach in the introduction on page 5, lines 122-126:

“We intentionally adopted a mixed design, combining both between-subjects and within-subject methodologies. The between-subjects approach was chosen to minimize the risk of carry-over effects and sequence biases. Simultaneously, we capitalized on the advantages of a within-subject design by altering the pre- to post-insight comparison and the mode of insight (imagination vs. observation) within each participant.”

Moreover, to provide a comprehensive portrayal of the two groups, we incorporated de-scriptions concerning trait and state variables alongside age and motor thresholds and in-cluded t-test comparisons between these variables on page 7, lines 157-160:

“Notably, the groups did not differ on levels of subjective chronic stress (TICS), state and trait anxiety (STAI-S, STAI-T), depressive mood (BDI), imaginative capacities (FFIS), person-ality dimensions (BFI), age, and motor thresholds (for descriptive statistics see Table 1; all p > 0.053).”

And further included age and motor thresholds as control variables in Table 1 on page 18, lines 402-404:

“Overall, levels of subjective chronic stress, anxiety, and depressive mood were relatively low and not different between groups. The groups did further not differ in terms of per-sonality traits, imagination capacity, age or motor thresholds (all p > 0.053; see Table 1).”

For greater precision in outlining our hypotheses, we specified these at the end of the in-troduction on pages 4-55, lines 107-118:

“Considering this involvement of the angular gyrus in imaginative processes, we expected that the effect of cTBS on the change in representational similarity from pre- to post-insight will differ based on the mode of insight – whether this insight was gained via imagination or observation. Specifically, we expected a more pronounced impairment in the neural recon-figurations when insight is gained via imagination, as this function may depend more on an-gular gyrus recruitment than insight gained via observation. Additionally, we expected cTBS to the left angular gyrus to interfere with the increase in neural similarity for linked events and with the decrease of neural similarity for non-linked events. We further predicted that cTBS to the left angular gyrus would reduce the impact of (imagination-based) insight into the link of initially unrelated events on memory performance during free recall, given its higher variability compared to other memory measures.”

1. Furthermore, in some instances, the types of analyses used do not appear to be suitable for addressing the questions posed by the current study, and there is limited explanation pro-vided for the choice of analyses and questionnaires.

We addressed this concern by inserting a new section “control variables” in the methods explaining our rationale for employing the different questionnaires as control var-iables on pages 40-41, lines 1003-1019:

“Control variablesIn order to ensure that the observed effects were solely attributable to the TMS manipula-tion and not influenced by other factors, we comprehensively evaluated several trait and state variables. To account for potential variations in anxiety levels that could impact our re-sults, we specifically measured state and trait anxiety using STAI-S and STAI-T (Laux et al., 1981), thus minimizing the potential confounding effects of anxiety on our findings (Char-pentier et al., 2021). Additionally, we evaluated participants’ chronic stress levels using the TICS (Schulz & Schlotz, 1999) to exclude any group variations that might explain the effect on memory, cosidering the well-established impact of stress on memory (Sandi & Pinelo-Nava, 2007; Schwabe et al., 2012). Moreover, we assessed participants’ depressive symp-toms employing the BDI (Hautzinger et al., 2006), to guarantee group comparability on this clinical measure. We further assessed fundamental personality dimensions using the BFI-2 (Danner et al., 2016) to exclude any potential group discrepancies that could account for dif-ferences observed. Lastly, we assessed participants’ imaginative capacities using the FFIS (Zabelina & Condon, 2019), to ensure uniformity across groups regarding this central varia-ble, considering the significant role of imagination in relation to the cTBS-targeted angular gyrus (Thakral et al., 2017).”

We further specified why we chose to analyze our behavioral data using LMMs on page 34, lines 849-85:

“For our behavioral analyses we opted to employ linear-mixed models (LMM), given their high robustness regarding the underlying distribution and high sensitivity to individual varia-tion (Pinheiro & Bates, 2000; Schielzeth et al., 2020).”

Moreover, we added an explanation on why we opted for the RSA approach in the meth-ods section on page 37, lines 920-923:

“This method is ideally suited to measure neural representation changes and was specifical-ly chosen as it has been previously identified as the preferred approach for quantifying in-sight-induced neural changes (Grob et al., 2023b; Milivojevic et al., 2015).”

To clarify on the rationale behind our coherence analysis, we incorporated an explanatory sentence in the methods section on page 39, lines 966-967:

“Due to the robust connectivity between the angular gyrus and other brain regions (Petit et al., 2023; Seghier, 2013), we proceeded with a connectivity analysis as a next step.”

**Reviewer 3**

We thank the reviewer for the constructive and very helpful comments. We are pleased that the reviewer considered our experimental design to be strong and our behavioral results to be striking.

1. My major criticism relates to the main claim of the paper regarding causality between the angular gyrus and the authors' behavior of interest. Specifically, I am not convinced by the evidence that the effects of stimulation noted in the paper are attributable specifically to the angular gyrus, and not other regions/networks.

While our results showed specific changes after cTBS over the angular gyrus, demonstrating a causal involvement of the angular gyrus in these effects, we completely agree that this does not rule out an involvement of additional areas. In particular, there is evidence suggesting that cTBS over parietal regions, such as the angular gyrus, could poten-tially influence hippocampal functioning. We address this issue now in a new paragraph that we have added to the discussion, on pages 23-24, lines 546-564:

“Another intriguing aspect to consider is that the stimulated site was situated in the more ventral portion of the angular gyrus, recognized for its stronger connectivity to the episodic hippocampal memory system in contrast to its more dorsal counterpart (Seghier, 2013; Ud-din et al., 2010). This stronger connectivity between the ventral angular gyrus and the hip-pocampus may shed light on the greater impact of cTBS to the angular gyrus on imagination-based insight. Given the angular gyrus’s robust connectivity with other brain regions, includ-ing the hippocampus (Seghier, 2013), it is plausible that the observed changes might not solely stem from alterations within the angular gyrus itself, but could also originate from these interconnected regions. This notion may bear particular importance given the re-quired accessibility to the hippocampus during imaginative processes (Benoit & Schacter, 2015; Grob et al., 2023a; Zeidman & Maguire, 2016). Interactions between the angular gyrus and the hippocampus may give rise to rich memory representations (Ramanan et al., 2018). In line with this, recent studies have demonstrated that cTBS to the angular gyrus resulted in enhanced hippocampal connectivity and improved associative memory (Hermiller et al., 2019; Tambini et al., 2018; Wang et al., 2014). However, it should be noted that our study detected impaired associative memory following cTBS to the angular gyrus. Expanding upon this idea, it is conceivable that targeting a more dorsal segment of the angular gyrus might exert a stronger influence on observation-based linking – an aspect that warrants future in-vestigations.”

Responses to reviewer recommendations

**Reviewer 1**
1. On page 26, the authors write: "[...] different video events (A, B, and X) were recalled from day one [...]". I may have missed this point, but I had the impression that the task was con-ducted within one day.

Indeed, this study was conducted within a single day. We rephrased the respec-tive statement accordingly. Please see page 7, lines 149-153:

“To test this hypothesis and the causal role of the angular gyrus in insight-related memory reconfigurations, we combined the life-like video-based narrative-insight task (NIT) with representational similarity analysis of EEG data and (double-blind) neuro-navigated TMS over the left angular gyrus in a comprehensive investigation within a single day.”

We further included this information in the methods section on page 27, lines 634-635:

“In total, the experiment took about 4.5 hours per participant and was completed within a single day. ”

**Reviewer 2**
1. There is a substantial disconnection between the introduction and the methods/results sec-tion. One reason is that there is not sufficient detail regarding the hypotheses/predictions and the specific types of analyses chosen to test these hypotheses/predictions. Additionally, it is not explained what comparisons and outcomes would be informative/expected. This should be made clear. Second and related to the above, the rationale for conducting certain types of analyses (correlation, coherence, see below) sometimes is not specified.

To address this concern, we elaborated on our hypotheses incorporating specif-ic predictions for the free recall, given its higher variability than the other memory measures, and for imagination vs. observation at the end of the introduction on pages 4-5, lines 107-122:

“Considering this involvement of the angular gyrus in imaginative processes, we expected that the effect of cTBS on the change in representational similarity from pre- to post-insight will differ based on the mode of insight – whether this insight was gained via imagination or observation. Specifically, we expected a more pronounced impairment in the neural recon-figurations when insight is gained via imagination, as this function may depend more on an-gular gyrus recruitment than insight gained via observation. Additionally, we expected cTBS to the left angular gyrus to interfere with the increase in neural similarity for linked events and with the decrease of neural similarity for non-linked events. We further predicted that cTBS to the left angular gyrus would reduce the impact of (imagination-based) insight into the link of initially unrelated events on memory performance during free recall, given its higher variability compared to other memory measures. Considering the high connectivity profile of the angular gyrus within the brain (Seghier, 2013), we conducted an EEG connec-tivity analysis building upon prior findings concerning alterations in neural reconfigurations. To establish a link between neural and behavioral findings, we chose a correlational ap-proach to relate observations from these two domains.”

Moreover, we made our rationale for the employed analyses more explicit and specified why we chose to analyze our behavioral data using LMMs on page 34, lines 849-851:

“For our behavioral analyses we opted to employ linear-mixed models (LMM), given their high robustness regarding the underlying distribution and high sensitivity to individual varia-tion (Pinheiro & Bates, 2000; Schielzeth et al., 2020).”

Moreover, we added an explanation on why we opted for the RSA approach in the meth-ods section on page 37, lines 920-923:

“This method is ideally suited to measure neural representation changes and was specifical-ly chosen as it has been previously identified as the preferred approach for quantifying in-sight-induced neural changes (Grob et al., 2023b; Milivojevic et al., 2015).”

To clarify on the rationale behind our coherence analysis, we incorporated an explanatory sentence in the methods section on page 39, lines 966-967:

“Due to the robust connectivity between the angular gyrus and other brain regions (Petit et al., 2023; Seghier, 2013), we proceeded with a connectivity analysis as a next step.”

1. The authors suggest that besides Branzi et al. (2021), this is one of the first studies showing that memory update is linked to the AG. I suggest having a look at work from Tambini, Nee, & D'Esposito, 2018, JoCN, and other papers from Joel Voss' group that target a similar re-gion of AG/Inferior parietal cortex. Many studies, using multiple TMS protocols, have now shown this brain region is causally involved in episodic and associative memory encoding.As mentioned above, further consideration of this literature is important as it delves into the region's hippocampal connectivity (and other network properties), and how that mediates the memory effects. Indeed because of the nature of the methods employed in this study, we do not know if the memory-related behavioural effects are due to TMS-changes induced at the AG's versus the hippocampal' s level, or both. How do the current findings square with the existing TMS effects from this region? Can the connectivity profile of the target re-gion highlighted by previous studies provide further insight into how the current behaviour-al effect arises? Some comments on this could be added to the discussion.

We completely agree that the other studies showing enhanced associative memory after TMS to parietal regions need to be addressed. Therefore, we updated the discussion on page 20, lines 449-453:

“Interestingly, recent work has additionally indicated that targeting parietal regions with TMS led to alterations in hippocampal functional connectivity, thereby enhancing associa-tive memory (Nilakantan et al., 2017; Tambini et al., 2018; Wang et al., 2014), potentially shedding light on the underlying mechanisms involved.”

Moreover, we included a section specifically addressing the possibility that the effects ob-served may pertain to having modulated other regions via the targeted region and updated the discussion on pages 23-24, lines 543-562:

“Furthermore, the differential impact of cTBS to the angular gyrus on neural reconfigura-tions between events linked via imagination and those linked via observation may be at-tributed to its crucial role in imaginative processes (Ramanan et al., 2018; Thakral et al., 2017). Another intriguing aspect to consider is that the stimulated site was situated in the more ventral portion of the angular gyrus, recognized for its stronger connectivity to the episodic hippocampal memory system in contrast to its more dorsal counterpart (Seghier, 2013; Uddin et al., 2010). This stronger connectivity between the ventral angular gyrus and the hippocampus may shed light on the greater impact of cTBS to the angular gyrus on im-agination-based insight. Given the angular gyrus’s robust connectivity with other brain re-gions, including the hippocampus (Seghier, 2013), it is plausible that the observed changes might not solely stem from alterations within the angular gyrus itself, but could also origi-nate from these interconnected regions. This notion may bear particular importance given the required accessibility to the hippocampus during imaginative processes (Benoit & Schacter, 2015; Grob et al., 2023a; Zeidman & Maguire, 2016). Interactions between the an-gular gyrus and the hippocampus may give rise to rich memory representations (Ramanan et al., 2018). In line with this, recent studies have demonstrated that cTBS to the angular gy-rus resulted in enhanced hippocampal connectivity and improved associative memory (Hermiller et al., 2019; Tambini et al., 2018; Wang et al., 2014). However, it should be noted that our study detected impaired associative memory following cTBS to the angular gyrus.”

1. Another comment I have regards the results observed for the observation vs imagination insight conditions. The authors mention that the 'changes in representational similarity for the observation condition should be interpreted with caution, as these seemingly opposite changes appeared to be at least in part driven by group differences already in the pre-phase before participants gained insight.' I wonder what these group differences are and whether the authors have any hypothesis about what factors determined them.

We could only speculate about the basis of the observed pre-insight phase dif-ferences. However, we provide now the raw RSA data as supplemental material to make the pattern of the (raw) RSA findings in the pre- and post-insight phases more transparent. We refer the interested reader to this material on pages 12-13, lines 293 to 295:

“For linked events, this analysis yielded a negative cluster (p = 0.032, ci-range = 0.00, SD = 0.00) in the parieto-temporal region (electrodes: T7, Tp7, P7; Fig. 3B; Figure 3 – Figure sup-plement 1).”

And on page 15, lines 339-341:

“This analysis yielded a positive cluster (p = 0.035, ci-range = 0.00, SD = 0.00) in a fronto-temporal region (electrode: FT7; Fig. 3C; Figure 3 – Figure supplement 2).”

Furthermore, the age of participants is not reported separately for the two groups (cTBS to AG vs Sham), I think. This should be reported including a t-test showing that the two groups have the same age.

We agree and report now explicitly that groups did not significantly differ in rel-evant control variables including age. Please see page 7, lines 157-160:

“Notably, the groups did not differ on levels of subjective chronic stress (TICS), state and trait anxiety (STAI-S, STAI-T), depressive mood (BDI), imaginative capacities (FFIS), person-ality dimensions (BFI), age, and motor thresholds (for descriptive statistics see Table 1; all p > 0.053).”

And further included age and motor thresholds as control variables in Table 1 on page 18, lines 402-412:

“Overall, levels of subjective chronic stress, anxiety, and depressive mood were relatively low and not different between groups. The groups did further not differ in terms of per-sonality traits, imagination capacity, age or motor thresholds (all p > 0.053; see Table 1).”

The fact this study is not a within-subject design makes difficult the interpretation of the results and this should be recognised as an important limitation of the study.

As outlined above, a within-subject design would in our view come with several disadvantages, such as significant sequence/carry-over effects. Moreover, the neural rep-resentation change was measured in a pre-post design, enabling us to measure the insight-driven neural reconfiguration at the individual level.

We clarify our rationale for the between-subjects factor TMS in the introduction on page 5, lines 122-126:

“We intentionally adopted a mixed design, combining both between-subjects and within-subject methodologies. The between-subjects approach was chosen to minimize the risk of carry-over effects and sequence biases. Simultaneously, we capitalized on the advantages of a within-subject design by altering the pre- to post-insight comparison and the mode of insight (imagination vs. observation) within each participant.”

Furthermore, we included our rationale for choosing a between-subjects approach for the crucial TMS manipulation in the methods section on page 25, lines 601-604:

“We implemented a mixed-design including the within-subject factors link (linked vs. non-linked events), session (pre- vs. post-link), and mode (imagination vs. observation) as well as the between-subjects factor group (cTBS to the angular gyrus vs. sham) to mitigate the risk of carry-over effects and sequence biases of the crucial cTBS manipulation.”

1. The angular gyrus is a heterogeneous region with multiple graded subregions. The one tar-geted in the present study is the ventral AG which has strong connections with the episodic-hippocampal memory system. I was wondering if this might explain why the AG TMS ef-fects on representational changes have been observed for events linked via imagination but not direct observation. Perhaps the stimulation of a more 'visual' AG subregion (see Hum-phreys et al., 2020, Cerebral Cortex) would have resulted in a different (opposite) pattern of results. It would be good to add some comments on this in the discussion.

We appreciate this interesting perspective offered regarding the potential out-comes of our study, particularly in relation to the activation of a more ventral sub region of the angular gyrus. We incorporated this idea into our discussion, alongside considerations regarding the potential effects of a more dorsal angular gyrus stimulation on observation-based linking. However, caution is warranted recognizing the inherent limitations posed by the precision of TMS manipulations, which is further underscored by our electric field simu-lations, utilizing a 10 mm radius. We included this section in the discussion on pages 23-24, lines 546-569:

“Another intriguing aspect to consider is that the stimulated site was situated in the more ventral portion of the angular gyrus, recognized for its stronger connectivity to the episodic hippocampal memory system in contrast to its more dorsal counterpart (Seghier, 2013; Ud-din et al., 2010). This stronger connectivity between the ventral angular gyrus and the hip-pocampus may shed light on the greater impact of cTBS to the angular gyrus on imagina-tion-based insight. Given the angular gyrus’s robust connectivity with other brain regions, including the hippocampus (Seghier, 2013), it is plausible that the observed changes might not solely stem from alterations within the angular gyrus itself, but could also originate from these interconnected regions. This notion may bear particular importance given the re-quired accessibility to the hippocampus during imaginative processes (Benoit & Schacter, 2015; Grob et al., 2023a; Zeidman & Maguire, 2016). Interactions between the angular gyrus and the hippocampus may give rise to rich memory representations (Ramanan et al., 2018). In line with this, recent studies have demonstrated that cTBS to the angular gyrus resulted in enhanced hippocampal connectivity and improved associative memory (Hermiller et al., 2019; Tambini et al., 2018; Wang et al., 2014). However, it should be noted that our study detected impaired associative memory following cTBS to the angular gyrus. Expanding upon this idea, it is conceivable that targeting a more dorsal segment of the angular gyrus might exert a stronger influence on observation-based linking – an aspect that warrants future in-vestigations. Yet, while acknowledging the functional heterogeneity within the angular gy-rus (Humphreys et al., 2020), pinpointing specific sub regions via TMS remains challenging due to its limited focal precision at the millimeter level (Deng et al., 2013; Thielscher & Kammer, 2004), as reinforced by our electric field simulations utilizing a 10 mm radius. Hence, drawing definitive conclusions regarding distinct angular gyrus sub regions requires future research employing rigorous checks to assess the focality of their stimulation.”

1. Regarding the methods section, I have the following specific queries. It is unclear what is the purpose of the coherence and correlation analyses (pages 35, 36). Could the authors pro-vide further clarification on this? These analyses seem not to be mentioned anywhere in the introduction. This should be clarified briefly in the introduction and then in the methods sec-tion. The same for the questionnaires (anxiety, stress, etc): It is unclear the reason for col-lecting this type of data. This should be clarified in the introduction as well.

We agree, and have updated the introduction as follows on page 5, lines 118-122:

“Considering the high connectivity profile of the angular gyrus within the brain (Seghier, 2013), we conducted an EEG connectivity analysis building upon findings from the RSA anal-yses concerning alterations in neural reconfigurations. To establish a link between neural and behavioral findings, we chose a correlational approach to relate observations from these two domains.”

We additionally provided an explanation for including these questionnaires in the introduc-tion on page 5, lines 126-129:

“To control for any group differences beyond the TMS manipulation, we gathered various control variables through questionnaires, including trait- and state-anxiety, depressive symptoms, chronic stress levels, personality dimensions, and imaginative capacities.”

Moreover, we elaborated on the underlying rationale guiding our chosen analytical ap-proaches. Therefore, we specified why we chose to analyze our behavioral data using LMMs on page 34, lines 849-851:

“For our behavioral analyses we opted to employ linear-mixed models (LMM), given their high robustness regarding the underlying distribution and high sensitivity to individual varia-tion (Pinheiro & Bates, 2000; Schielzeth et al., 2020).”

Furthermore, we added an explanation on why we opted for the RSA approach in the methods section on page 37, lines 920-923:

“This method is ideally suited to measure neural representation changes and was specifical-ly chosen as it has been previously identified as the preferred approach for quantifying in-sight-induced neural changes (Grob et al., 2023b; Milivojevic et al., 2015).”

To clarify on the rationale behind our coherence analysis, we incorporated an explanatory sentence in the methods section on page 39, lines 966-967:

“Due to the robust connectivity between the angular gyrus and other brain regions (Petit et al., 2023; Seghier, 2013), we proceeded with a connectivity analysis as a next step.”

1. The preregistration webpage is in German. This is not ideal as it means that the information is available only to German speakers.

This webpage can easily be switched to English by changing the settings in the top right corner:

To address this issue, we included a description of how to set the webpage to English in the methods section on page 25, lines 581-582:

“For translation to English, please adjust the page settings located in the top right corner.”

1. Page 18. 'NIT' and 'MAT' - avoid abbreviations when possible.

We included the full name for the narrative-insight task (NIT) on page 7, line 151, line 153, and line 165, page 8 lines 177-178 and line 187, page 19 on line 427, page 26 on line 615, line 629 and line 632, page 27, line 653, page 30, lines 730-731, page 31, line 754, page 35, line 870, line 873, and page 36 and line 885.

We further included the full name for the multi-arrangements task (MAT) on page 19, lines 428-429.

1. Line 21....we further observed DECREASED...should be replaced with INCREASED, if I am not wrong.

We checked the sentence again and it looks correct to us, since it describes the change for observation-based insight, not imagination-based insight. We clarified that this finding pertains to observation-based linking by modifying the sentence on page 23, lines 525-528, as follows:

“Following cTBS to the angular gyrus, we further observed decreased pattern similarity for non-linked events in the observation-based condition, resembling the pattern change ob-served in the sham group for linked events, which may highlight the role of the angular gy-rus in representational separation during observation-based linking”

**Reviewer 3**
1. The major claim of the paper is that the angular gyrus is causally involved in insight-driven memory reconfiguration. To the authors' credit, they localized stimulation to the angular gyrus using an anatomical scan, the strength of the estimated electromagnetic field in the angular gyrus correlated with their behavioral results, and there were also brain-behavior correlations involving sensors located in the parietal lobe. However, the minimum evidence needed to claim causality is (1) evidence of a behavioral change (which the authors found) and (2) evidence of target engagement in the angular gyrus. It is also important to show brain-behavior correlations between target engagement and behavior. Although the au-thors stimulated the angular gyrus, that does not mean that rTMS specifically affected this region or that the behavioral results can be attributed to rTMS effects on the angular gyrus. As the authors point out, the angular gyrus has dense connections with other regions such as the hippocampus. In fact, several studies have shown that angular gyrus (or near AG) stimulation affects the hippocampal network (Wang et al., 2014, Science; Freedberg et al. 2019, eNeuro; Thakral et al., 2020, PNAS). EEG also has a poor spatial resolution, so even though the results were attributable to parieto-temporal sensors, this is not sufficient evi-dence to claim that the angular gyrus was modulated. Source localization would be re-quired to reconstruct the signal specifically from the AG. Thus, with the manuscript written as is, the authors can claim that "cTBS to the angular gyrus modulates insight-driven memory reconfiguration," but the current claim is not sufficiently substantiated.

While acknowledging the potential role of the angular gyrus in driving the ob-served changes, we recognize that the available evidence may not be sufficient. Conse-quently, we have introduced several modifications within our manuscript to address this concern.

In the revised Introduction, we now explicitly address the possibility of a stimulation of the hippocampus via the angular gyrus on page 4, lines 84-85:

“Accordingly, previous studies have shown that stimulation of the angular gyrus resulted in altered hippocampal activity (Thakral et al., 2020; Wang et al., 2014).”

Additionally, we included relevant evidence demonstrating previous instances of targeted stimulation of the angular gyrus, which led to alterations in hippocampal connectivity and associative memory. These insights have been included in the discussion on page 20, lines 449-453:

“Interestingly, recent work has additionally indicated that targeting parietal regions with TMS led to alterations in hippocampal functional connectivity, thereby enhancing associa-tive memory (Nilakantan et al., 2017; Tambini et al., 2018; Wang et al., 2014), potentially shedding light on the underlying mechanisms involved.”

Next, we have integrated crucial modifications essential for establishing a conclusive infer-ence of causality in our study. Moreover, we now explore the potential mediation of the effects observed from angular gyrus stimulation through other brain regions, like the hip-pocampus. In addition, we have highlighted prior work where such stimulation coincided with alterations in associative memory. For the updated discussion section, please see pag-es 23-24, lines 538-562:

“Although our study provided evidence suggesting a causal role of the angular gyrus in in-sight-driven memory reconfigurations – highlighted by behavioral changes after cTBS to the angular gyrus, neural changes in left parietal regions, and relevant brain-behavior associa-tions – it is important to acknowledge the limitations imposed by the spatial resolution of EEG. Consequently, the precise source of the observed signal changes in the parietal re-gions remains uncertain, potentially tempering the definitive nature of these findings. Fur-thermore, the differential impact of cTBS to the angular gyrus on neural reconfigurations between events linked via imagination and those linked via observation may be attributed to its crucial role in imaginative processes (Ramanan et al., 2018; Thakral et al., 2017). An-other intriguing aspect to consider is that the stimulated site was situated in the more ven-tral portion of the angular gyrus, recognized for its stronger connectivity to the episodic hippocampal memory system in contrast to its more dorsal counterpart (Seghier, 2013; Ud-din et al., 2010). This stronger connectivity between the ventral angular gyrus and the hip-pocampus may shed light on the greater impact of cTBS to the angular gyrus on imagina-tion-based insight. Given the angular gyrus’s robust connectivity with other brain regions, including the hippocampus (Seghier, 2013), it is plausible that the observed changes might not solely stem from alterations within the angular gyrus itself, but could also originate from these interconnected regions. This notion may bear particular importance given the re-quired accessibility to the hippocampus during imaginative processes (Benoit & Schacter, 2015; Grob et al., 2023a; Zeidman & Maguire, 2016). Interactions between the angular gyrus and the hippocampus may give rise to rich memory representations (Ramanan et al., 2018). In line with this, recent studies have demonstrated that cTBS to the angular gyrus resulted in enhanced hippocampal connectivity and improved associative memory (Hermiller et al., 2019; Tambini et al., 2018; Wang et al., 2014). However, it should be noted that our study detected impaired associative memory following cTBS to the angular gyrus.”

We further replaced terms that imply inhibition of the angular gyrus with a more operation-ally descriptive phrase:

“cTBS to the angular gyrus”

1. The authors frequently claim that cTBS is "inhibitory stimulation" and that inhibition of the angular gyrus caused their effects. There is a common misconception within the cognitive neuroscience literature that stimulation is either "inhibitory" or "excitatory," but there is no such thing as either. The effects of rTMS are dependent on many physiological, state, and trait-specific variables and the location of stimulation. For example, while cTBS does repro-ducibly inhibit behavior supported by the motor cortex (Wilkinson et al., 2010, Cortex; Rosenthal et al., 2009, J Neurosci), cTBS of the posterior parietal cortex reproducibly en-hances hippocampal network functional connectivity and episodic memory (Hermiller et al., 2019, Hippocampus; Hermiller et al., 2020, J Neurosci). The authors reference the Huang et al. (2005) paper as evidence of its inhibitory effects but work in this paper is not sufficient to broadly categorize cTBS as inhibitory. First, Huang et al. stimulated the motor cortex and measured the effects on corticospinal excitability, which is significantly different from what the current authors are measuring. Furthermore, this oft-cited study only included 9 sub-jects. Other studies have found that the effects of theta-burst are significantly more varia-ble when more subjects are used. For example, intermittent theta-burst, which is assumed to be excitatory based on the Huang paper, was found to produce unreliable excitatory ef-fects when more subjects were examined (Lopez-Alonso, 2014, Brain Stimulation). Thus, the a priori assumption that stimulation would be inhibitory is weak and cTBS should not be dis-cussed as "inhibitory."

We agree and included now a statement in the methods section that explicitly states that cTBS effects may be region-specific on page 33, lines 817-819:

“Nonetheless, the effects of cTBS appear to vary based on the targeted region, with cTBS to parietal regions demonstrating the capability to enhance hippocampal connectivity (Hermiller et al., 2019, 2020).”

We further substituted all terminology suggestive of an inhibitory effect with the phrase:

“cTBS to the angular gyrus”.

However, it is important to note, that while other studies (Hermiller et al., 2019; Tambini et al., 2018; Wang et al., 2014) found increased hippocampal connectivity after rTMS to a parie-tal region as well as enhanced associative memory, we observed impaired memory for the linked events. We included this clarification in the discussion on page 24, lines 558-562:

“In line with this, recent studies have demonstrated that cTBS to the angular gyrus resulted in enhanced hippocampal connectivity and improved associative memory (Hermiller et al., 2019; Tambini et al., 2018; Wang et al., 2014). However, it should be noted that our study detected impaired associative memory following cTBS to the angular gyrus.”

1. The hypothesis at the end of the introduction did not strike me as entirely clear. From this hypothesis, it seems that the authors are just comparing the differences in memory and re-configuration during imagination-based insight links. However, the authors also include ob-servation-based links and a non-linking condition, which seem ancillary to the main hy-pothesis. Thus, I am confused about why these extra factors were included and exactly what statistical results would confirm the authors' hypothesis.

We agree, and have clarified our hypotheses on pages 4-5, lines 107-115:

“Considering this involvement of the angular gyrus in imaginative processes, we expected that the effect of cTBS on the change in representational similarity from pre- to post-insight will differ based on the mode of insight – whether this insight was gained via imagination or observation. Specifically, we expected a more pronounced impairment in the neural recon-figurations when insight is gained via imagination, as this function may depend more on an-gular gyrus recruitment than insight gained via observation. Additionally, we expected cTBS to the left angular gyrus to reduce the increase in neural similarity for linked events and in-crease of neural dissimilarity for non-linked events.”

1. Many of the distributions throughout the paper do not look normal. Was normality checked? Are non-parametric stats warranted?

We evaluated and reported the normality assumption in our behavioral anal-yses. Despite the non-normal distribution of our data, we chose to utilize linear-mixed models due to their robust performance even in case of deviations from normal distribu-tions. This update in our methods section can be found on page 36, lines 890-896:

“After outlier correction, we identified non-normality in our data using a Shapiro-Wilk test (narrative-insight task: W = 0.92, p < 0.001; multi-arrangements task: W = 0.94, p < 0.001; forced-choice recognition: W = 0.50, p < 0.001; free recall details: W = 0.85, p < 0.001; free recall naming of linking events: W = 0.94, p < 0.001). However, we mitigated this by employ-ing linear-mixed models (LMMs), recognized for their robustness even with non-normally distributed data (Schielzeth et al., 2020).”

We recalculated the correlational analysis between the RSA data and the behavioral recall of linking events by using the Spearman method on page 13, lines 306-308:

“Furthermore, to address a deviation from the normality assumption, the correlational analysis was repeated using the Spearman method, which indicated an even stronger cor-relation (r(59) = 0.32, p = 0.012).”

We further recalculated the correlation between the change in coherence for linked events and the recall of details for events linked via imagination on page 16, lines 376-378:

“Please note that for addressing a deviation from the normality assumption, the correla-tional analysis was repeated using the Spearman method, which yielded a significant corre-lation of similar strength (r(59) = 0.31, p = 0.015).”

Our EEG analyses , including RSA and coherence analyses, utilized a cluster-based permuta-tion test (Fieldtrip; Oostenveld et al., 2011). These tests do not assume a normal distribu-tion by utilizing empirical sampling for statistical inference. This approach ensures robust-ness without constraints imposed by specific distributional assumptions. Subsequent t-tests, stemming from significant clusters identified in the initial non-parametric analyses, were extensions of the robust non-parametric approach and did not require additional normality testing.

1. Can the authors include more detail about the sham coil? Was it subthreshold? Did the EMF cross the skull?

The sham coil, also obtained from MAG & More GmbH, München, Germany, provided a similar sensory experience; however, the company did not specify any field strength (n.a.) as this coil was purposefully designed to prevent the induction of an elec-tromagnetic field (EMF) capable of penetrating the skull, thereby ensuring it had no impact on the brain. We clarified on this point in the methods section on pages 31-32, lines 772-778:

“Two identically looking but different 70 mm figure-of-eight-shaped coils were used de-pending on the TMS condition: The PMD70-pCool coil (MAG & More GmbH, München, Germany) with a 2T maximum field strength was used for cTBS, while the PMD70-pCool-SHAM coil (MAG & More GmbH, München, Germany), with minimal magnetic field strength, was employed for sham, providing a similar sensory experience, with stimulation pulses being scattered over the scalp and not penetrating the skull.”

1. There are differences between exclusion criteria in pre-registration and report. For example, BMI is an exclusion factor in the report, but not in the pre-registration. Can the authors provide a reason for this deviation?

This discrepancy is due to (partial) participant recruitment from previous fMRI studies conducted in our lab that involved a stress induction protocol (as a structural MRI image was needed for the ‘neuronavigated’ TMS). Owing to the distinct cortisol stress reac-tivity observed in individuals with varying body mass indices (BMIs), participants with a BMI below 19 or above 26 kg/m² were excluded from these studies. To maintain consistency within our sample, only participants meeting these criteria were included. We elaborated on this point in the methods section on page 25, lines 586-592:

“Participants were screened using a standardized interview for exclusion criteria that com-prised a history of neurological and psychiatric disease, medication use and substance abuse, cardiovascular, thyroid, or renal disease, evidence of COVID-19 infection or expo-sure, and any contraindications to MRI examination or TMS. Additionally, participants with a body mass index (BMI) below 19 or above 26 kg/m² were excluded. This decision stemmed from recruiting some participants from prior studies that incorporated stress induction pro-tocols, which imposed this specific criterion (Herhaus & Petrowski, 2018; Schmalbach et al., 2020).”

1. Were impedances monitored and minimized during EEG?

Yes, they were monitored. We clarified this point in the methods section on page 34, lines 845-847:

“We maintained impedances within a range of ± 20 μV using the common mode sense (CMS) and driven right leg (DRL) electrodes, serving as active reference and ground, re-spectively”

1. I think there may be a typo related to the Thakral coordinates. I believe Thakral used MNI coordinates -48,-64, 30, whereas the authors stated they used -48,-67,30. Is this a mistake?

Upon reevaluation of our study coordinates, we identified a slight deviation in our stimulation coordinates compared to those reported by Thakral et al. (2017; +3mm on the y-axis). This variance resulted from the required MNI to Talairach (TAL) transformations necessary for utilizing the neuronavigation software Powermag View! (MAG & More GmbH, München, Germany). Notably, this deviation was consistent across all participants in our study. While TMS is more precise than tDCS, its focality is not as fine-grained down to the millimeter level. Despite this, our electric field simulations, adopting a 10mm radius, ef-fectively encompassed the original coordinates specified by Thakral et al. (2017). This radius ensured coverage over the intended target area, mitigating the impact of this minor devia-tion on the overall study outcomes. We updated the methods section accordingly on page 33, lines 800-806:

“Based on the individual T1 MR images, we created 3D reconstructions of the participants' heads, allowing us to precisely locate the left angular gyrus coordinate (MNI: -48, -67, 30), initially derived from previous work (Thakral et al., 2017), for TMS stimulation. Despite a mi-nor deviation in coordinates due to necessary MNI to Talairach transformations for soft-ware compatibility (Powermag View! by MAG & More GmbH, München, Germany), our methodology ensured precise localization of the angular gyrus target area.”

1. How was the tail of the coil positioned during stimulation? Was it individualized so that the lobes of the coil are perpendicular to the nearest gyrus, as is commonly done?

The coil handle always pointed upwards to maintain optimal positioning with the coil holder. We followed the positioning procedure in the neuronavigation software Powermag View!, which did not indicate any positioning of the coil handle but specified the position and angle of the coil itself. To incorporate this aspect, we updated the legend of figure 2 on page 11, lines 260-261:

“Please note that in the study, the coil handle was oriented upwards; however, in this illus-tration, it has been intentionally depicted as pointing downwards for better visibility pur-poses.”

We further updated the method section on page 33, lines 723-824:

“The coil was positioned tangentially on the head and mechanically fixed in a coil holder, with its handle pointing upwards to maintain its position”